# UNSUPERVISED HIERARCHICAL VIDEO PREDICTION

## ABSTRACT

Much recent research has been devoted to video prediction and generation, yet a lot of previous work has been focused on short-scale time horizons. The hierarchical video prediction method by Villegas et al. (2017) is an example of a state of the art method for long term video prediction. However, their method has limited applicability in practical settings as it requires a ground truth pose (e.g., poses of joints of a human) at training time. This paper presents a long term hierarchical video prediction model that does not have such a restriction. We show that the network learns its own higher level structure (e.g., pose-equivalent hidden variables) that works better in cases where the ground truth pose does not fully capture all of the information needed to predict the next frame. This method gives sharper results than other video prediction methods which do not require a ground truth pose, and its efficiency is shown on the Humans 3.6M and Robot Pushing datasets.

## 1 INTRODUCTION

It is hypothesized that learning to predict the future and the effect of their actions is an important quality for intelligent agents that interact with their environment. This is a complicated task, as typical use cases require predicting the outcome of interactions between the agent and objects over multiple timesteps.

In this work we are looking at the task of predicting the pixels of future video frames given the first few observed frames. We also consider the action conditional setting, in which we are given the action that the agent is taking and are tasked to predict the pixel level outcome of that action in the future.

The method of Villegas et al. (2017) is a novel way to generate long term video predictions, but requires ground truth human pose annotations. In this work we explore ways to generate videos using a hierarchical model *without* requiring a ground truth pose or other high level structure annotations for each frame. The method is hierarchical in the sense that it learns to generate a high level structure, then makes next frame predictions based on that structure.

## 2 RELATED WORK

**Patch level prediction**   The video prediction problem was initially studied at the patch level (Sutskever et al., 2009; Michalski et al., 2014; Mittelman et al., 2014; Srivastava et al., 2015). This work showed promising results on synthetic data (e.g. bouncing balls), but did not scale to predicting higher resolution videos.

**Frame level prediction on realistic videos.**   More recently, the video prediction problem has been formulated at the entire frame level. Most of the recent work is based on the convolutional encoder/decoder framework. Finn et al. (2016) proposed a network that can perform next level video frame prediction by explicitly predicting movement. For each pixel in the previous frame, the network outputs a distribution over locations that pixel is predicted to move. The movements are averaged to get the final prediction. The network is trained end to end to minimize L2 loss. Mathieu et al. (2016) proposed adversarial training with multiscale convolutional networks to generate sharper pixel level predictions in comparison to conventional L2 loss. Villegas et al. (2017) proposed a network that decomposes motion and content in video prediction and showed improved

performance over Mathieu et al. (2016). Lotter et al. (2017) proposed a deep predictive coding network in which each layer learns to predict the lower-level difference between the future frame and current frame. As an alternative approach to convolutional encoder/decoder networks, Kalchbrenner et al. (2016) proposed an autoregressive generation scheme for improved prediction performance. Despite their promise, these work have not been demonstrated for long term prediction on high resolution natural videos beyond $\approx$ 20 frames.

**Long-term prediction.** Oh et al. (2015) proposed action conditional convolutional encoder-decoder architecture that has demonstrated impressive long-term prediction performance on video games (e.g., Atari games), but it has not been applied for predicting challenging real-world videos.

## 2.1 HIERARCHICAL VIDEO PREDICTION (VILLEGAS ET AL., 2017)

Villegas et al. (2017) demonstrated a long-term prediction method using hierarchical prediction where the ground truth human pose is assumed to be given as supervision. Our method is based off of that work, so we describe it in detail in the following section.

### 2.1.1 INFERENCE AND ARCHITECTURE

To generate the image at timestep $t$, the following procedure is used. First, a convolutional neural network encoder generates an embedding vector from the previous ground truth image: $e_{t-1} = CNN(img_{t-1})$. This encoding represents the pose of a person.

Next, a multilayer LSTM predictor network predicts what the encoding will be in a future timestep. For some number of context frames, the predictor makes its prediction based off of the encoding from the ground truth image. After the predictor network has enough context, it makes its predictions based off of its previous predictions (Fig. 1 provides a helpful visual). For example, if there are C context frames, the following is used to generate the encoding at step $t$.

$$\begin{cases} [p_t, H_t] = LSTM(e_{t-1}, H_{t-1}) & if\ t <= C \\ [p_t, H_t] = LSTM(p_{t-1}, H_{t-1}) & if\ t > C \end{cases} \quad (1)$$

$H_t$ is the hidden state of the LSTM at timestep $t$. Note that only the encoding of the context frames are used, not the subsequent frames. Similar to $e_t$ in the above, $p_t$ represents the $predicted$ pose.

Once $p_t$ is obtained, a visual analogy network (VAN) (Reed et al., 2015) is used to generate the corresponding image at time $t$. The VAN applies the transformation that occurred between two images to a given query image. In this case the first frame of the video should be transformed in the same way as the encoding was transformed from the first to $t$-th timestep. The VAN does this by mapping images to a space where analogies can be represented by additions and subtractions, and then mapping the result back to image space. To obtain the predicted image at timestep $t$ using the VAN one needs to use $\widehat{img}_t = VAN(e_1, p_t, img_1)$, where the VAN is defined as

$$VAN(e_1, p_t, img_1) = f_{dec}(f_{enc}(g(p_t)) - f_{enc}(g(e_1)) + f_{img}(img_1)) \quad (2)$$

Where $g$ is a hardcoded function to transform the pose into a 2 dimensional representation of the pose. The weights of $f_{enc}$ and $f_{img}$ are shared.

### 2.1.2 TRAINING

The disadvantage of this method is that the training relies on ground truth pose annotations. The encoder is trained to produce the pose given the image, the predictor is trained to predict that pose into the future and the VAN is trained to generate the image given the pose.

## 3 PROPOSED METHOD

Our method uses a similar network architecture to Villegas et al. (2017) but we present ways of training the network that do not require a ground truth pose. In our method, $e_t$ and $p_t$ have the same

dimensionality and represent the network's own higher level structure (e.g., pose equivalent hidden variables) which the network learns as it is trained.

In our case, there is no straightforward way to transform the encoding into a 2 dimensional representation of the pose. Therefore, the part of the VAN that maps the encoding is a fully connected network instead of a convolutional neural network. As a result, the weights are not shared between the fully connected network which processes the encoding, and the ConvNet which processes the image.

The equation for the VAN becomes:

$$VAN(e_1, p_t, img_1) = f_{dec}(f_{enc}(p_t) - f_{enc}(e_1) + f_{img}(img_1)) \tag{3}$$

Note that $f_{enc}$ is a fully connected network, and $f_{img}$ is a conv net. $f_{dec}$ is a deconv network.

### 3.1 TRAINING

There are several ways these networks can be trained. In Villegas et al. (2017), they are each trained separately with the ground truth human pose. In this work, we explore alternative ways of training these networks in the absence of any ground truth pose or other high level structure annotations. We use the same procedure as Villegas et al. (2017) at inference time.

#### 3.1.1 END TO END

One option is to connect the networks the same way as in inference time and train them end to end (E2E). In this method, the L2 loss of of the predicted image is optimized: $\min(\sum_{t=1}^{T} L_2(\widehat{img}_t, img_t))$.

There are no constraints on what kind of encoding the encoder produces, or what kind of predictions the predictor makes. Because of how the networks are connected, the encoder will produce an encoding whose future state is easily predicted by the predictor. Likewise, the predictor will make predictions which the VAN can use to produce images which are similar to the ground truth. The encoder and predictor will not have to represent information that is present in the first ground truth frame, since the VAN will have access to the first frame. The size of $e_t$ and $p_t$ is a hyper parameter of this approach. Figure 1 represents a diagram of this method.

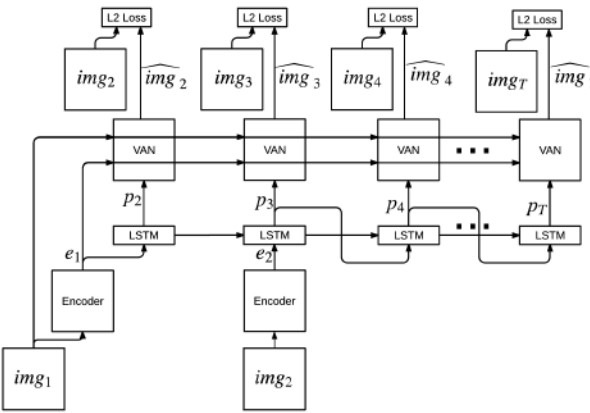

Figure 1: The E2E method. The first few frames are encoded and fed into the predictor as context. The predictor predicts the subsequent encodings, which the VAN uses to produce the pixel level predictions. The average of the losses is minimized. This is also the configuration of every method at inference time, even if the predictor and VAN are trained separately.

### 3.1.2 ENCODER PREDICTOR WITH ENCODER VAN

An alternative way to train the combined network is to explicitly train the encoder so that $e_t$ is easy to predict into the future, and so that the VAN can use $e_t$ to produce the next frame. We call this method Encoder Predictor with Encoder VAN, or EPEV. The encoder and predictor are trained together so the $e_t$ is easy to predict and the predictor predicts that encoding into the future. To accomplish this, the difference between $e_t$ and $p_t$, $L_2(e_t, p_t)$ is minimized. The encoder is also trained with the VAN so the VAN can use $e_t$ to produce the image and so that the encoder generates an informative encoding. This is done by minimizing the loss of the VAN given the encoder output: $L_2(\widehat{img}_{e_t}, img_t)$ where $\widehat{img}_{e_t} = VAN(e_1, e_t, img_1)$. The network is trained to minimize the sum of these two losses: $\min(\sum_{t=1}^{T} L_2(\widehat{img}_{e_t}, img_t) + \alpha L_2(e_t, p_t))$, where $\alpha$ is a hyper-parameter that controls the degree to which the $e_t$ will be easy to predict vs. informative enough so the VAN can produce a good image.

See figure 2 for a diagram of the encoder and predictor trained together, and figure 3 for the encoder and VAN trained together.

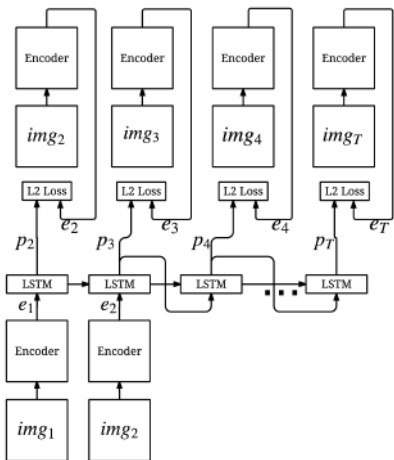

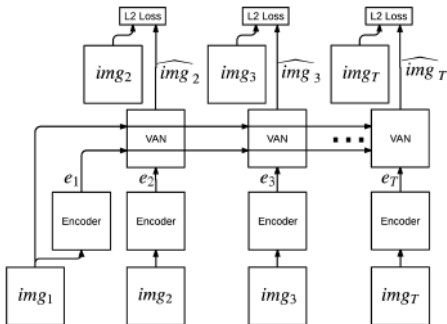

Figure 2: The segment of the EPEV method in which the encoder and predictor are trained together. The encoder is trained to produce an encoding that is easy to predict, and the predictor is trained to predict that encoding into the future. The average of the losses is minimized.

Figure 3: The segment of the EPEV method in which the encoder and VAN are trained together. The encoder is trained to produce an encoding that is informative to the VAN, while the VAN is trained to output the image given the encoding. The average of the losses is minimized. This method is similar to an autoencoder.

Separate gradient descent procedures (or optimizers, in TensorFlow parlance) could be used to minimize $L_2(\widehat{img}_{e_t}, img_t)$ and $L_2(e_t, p_t)$, but we find that minimizing the sum works better experimentally.

With this method, the predictor will predict the encoder outputs in future timesteps, and the VAN will use the encoder output to produce the frame.

### 3.1.3 E2E WITH POSE

The end to end approach can also be augmented if the dataset has information about the ground truth pose or any other high level frame annotations. In this method, the $e_t$ and $p_t$ vectors would be split into two: the first path is optimized to represent the pose, and the rest of $e_t$ and $p_t$ is trained the same way as the E2E approach. At each training step a separate optimizer minimizes each loss. In this method, we can think of $e_t$ and $p_t$ as the concatenation of two vectors, one representing the pose, and the other containing additional information the network can represent. If $e_t = [e_{pose_t}, e_{remaining_t}]$ and $p_t = [p_{pose_t}, p_{remaining_t}]$, the following losses are minimized:

The loss representing how well the encoder infers the pose: $\min(\sum_{t=1}^{T} L_2(e_{pose_t}, pose_t))$. The loss representing how well the predictor predicts the pose: $\min(\sum_{t=1}^{T} L_2(p_{pose_t}, pose_t))$. The end to end loss: $\min(\sum_{t=1}^{T} L_2(\widehat{img}_t, img_t))$.

These losses are minimized with separate optimizers in this method. Minimizing the end to end loss ensures that the VAN will learn to use the pose provided by the predictor network, and that the encoder and predictor will learn to produce additional information besides the pose that is useful to the VAN.

### 3.1.4 INDIVIDUAL

In order to compare to a baseline, we also implemented the method where each of the networks are trained individually, as in Villegas et al. (2017). The main difference between this method and Villegas et al. (2017) is that we do not use an adversarial loss (Goodfellow et al., 2014). See section 5 for a discussion of how an adversarial loss could be added to our method.

## 4 EXPERIMENTS

These methods were tested on two different datasets, the **Robot Push** dataset (Finn et al., 2016) and the **Humans 3.6M** dataset (Ionescu et al., 2014; Catalin Ionescu, 2011). Videos of the results of our method are available by visiting the following URL: `https://goo.gl/WA8uxc`.

The EPEV method works best experimentally if $\alpha$ starts small, around 1e-7, and is gradually increased to around .1 during training. As a result, the encoder will first be optimized to produce an informative encoding, then gradually optimized to also make that encoding easy to predict.

### 4.1 ROBOT PUSH DATASET

This dataset contains videos of a robot arm pushing objects on a table. The current joint angles and the location of the end effector are given, and we use these as the pose for the methods which require it. The action the robot arm is taking is fed into the predictor.

Each of the methods considered was given two frames of context, and then trained to predict 17 subsequent frames. An encoding size of 16 was used for the E2E method. The size of the pose is 12, so the encoding size of the INDIVIDUAL method is 12. The other methods used an encoding size of 32.

Additionaly, we randomly split the dataset into training, validation and test. We used 64x64 images, and the same frame rate as the original dataset. Results from our test set are shown in this section. Note that our experimental protocol is different from Finn et al. (2016), where the test set is composed of novel objects.

We hypothesized that the methods where the network could learn its own pose equivalent would predict the movement of the objects the robot arm pushes more accurately than the INDIVIDUAL method. To test this, we manually compared the E2E and EPEV methods to the INDIVIDUAL method and evaluated where the movement of predicted objects most closely matched the ground truth. We evaluated 40 videos in which objects move. The results are in Table 1.

Table 1: Results from manual comparison of object predictions in 40 videos. The methods perform similarly in the remaining videos.

| Comparison | Number of videos |
|---|---|
| EPEV better than INDIVIDUAL | 6 |
| INDIVIDUAL better than EPEV | 3 |
| E2E better than INDIVIDUAL | 6 |
| INDIVIDUAL better than E2E | 6 |

In the INDIVIDUAL method, the predictor network can only produce the pose, so the VAN has to infer how the objects will move based on the start and end state of the arm. We were surprised by how well the VAN could infer this. However, from examining the videos, the EPEV method had better

object predictions than the INDIVIDUAL method, which supports our hypothesis. The magnified part of ground truth frame 19 in figure 4 shows that the robot arm pushed the yellow object. The EPEV and E2E methods correctly predict this, but in the INDIVIDUAL method, the robot arm covers up the yellow object instead of moving it. Additional analysis is in appendix section F.

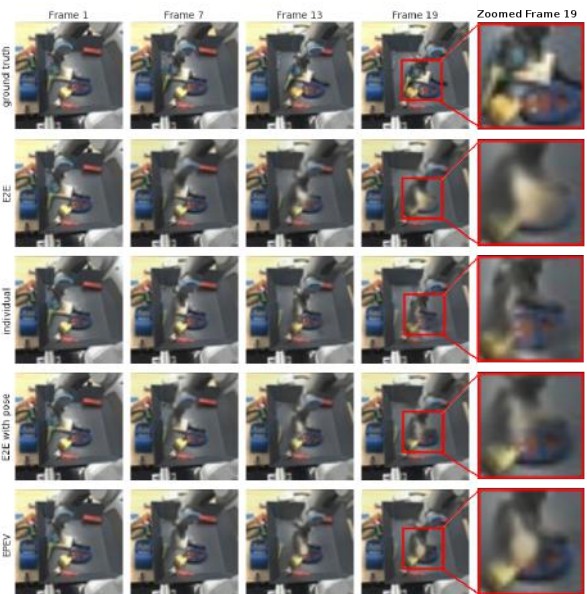

Figure 4: A visual comparison of the different methods on the robot push dataset. In the E2E and EPEV methods, the yellow object moves, but it does not in the INDIVIDUAL method.

The average Peak Signal to Noise Ratio (PSNR) of the different methods we introduce are similar on this dataset. In this data set, the model from Finn et al. (2016) gets a better PSNR than our model. The movement in this dataset can easily be represented by the movement of pixels, and it is relatively deterministic. So the model from Finn et al. (2016) which explicitly predicts the movement of objects and directly minimizes the L2 loss works well here.

## 4.2 LONG TERM PREDICTION ON A TOY DATASET

To confirm our claim that our method works well for long term predictions, we trained our method on a toy task with known factors of variation. We used a dataset with a generated shape that bounces around the image and changes size deterministically. We trained the EPEV method and the CDNA method in Finn et al. (2016) to predict 16 frames, given the first 3 frames as context. We do not show the E2E method since it usually predicts blurrier images than the EPEV method. Both methods are evaluated on predicting approximately 1k frames. We added noise to the LSTM states of the predictor network during training to help predict reasonable motion further into the future. Results form a held out test set are described in the following.

After visually inspecting the results of both methods, we found that when the CDNA fails, the shape disappears entirely, however when the EPEV method fails, the shape changes color. To quantitatively evaluate both methods, we used a script to measure whether a shape was present frames 1012 to 1022, and if that shape has the appropriate color. See table 2 for the results averaged over 1k runs. The CDNA method predicts a shape with the correct color about 25% of the time, and the EPEV method predicts a shape with the correct color about 97% of the time. The EPEV method sometimes fails by predicting the shape in the same location from frame to frame. This does not happen very often, as the reader can confirm by examining the randomly sampled predictions in appendix section E. It is unrealistic to expect the methods to predict the location of the shape accurately in frame 1000, since small errors propagate in each prediction step.

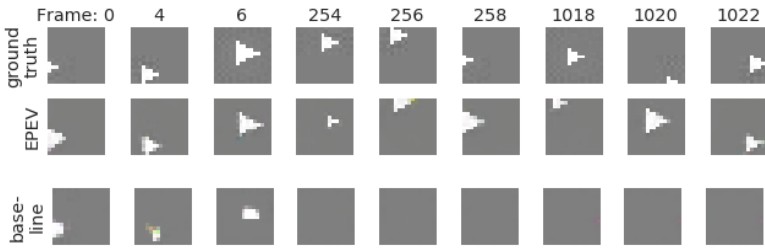

Figure 5: A visual comparison of the EPEV method and CDNA from Finn et al. (2016) as the baseline. This example is cherry picked to show the typical quality of predictions from both methods. See appendix section E for non-cherry picked results.

Table 2: Results on shapes dataset

| Method | Shape has correct color | Shape has wrong color | Shape disappeared |
|---|---|---|---|
| EPEV | 96.9% | 3.1% | 0% |
| CDNA Baseline | 24.6% | 5.7% | 69.7% |

## 4.3 HUMANS 3.6M

Our method was also tested on the Humans 3.6M Dataset. Only the E2E and EPEV methods were tested here, since Villegas et al. (2017) has already shown the results using the ground truth pose.

We used subjects 1, 5, 6, 7 and 8 for training, subject 9 for validation. Subject 11 results are reported in this paper for testing. We used 64 by 64 images. We subsampled the dataset to 6.25 frames per second. We trained the methods to predict 32 frames and the results in this paper show predicting 64 frames. Each method is given the first 5 frames as context frames. So the in these images, the model predicts about 10 seconds into the future from .8 seconds of context. We used an encoding size of 32 for the E2E method and a encoding size of 64 for the EPEV method on this dataset. We compare our method to the CDNA method in Finn et al. (2016) in Fig. 6.

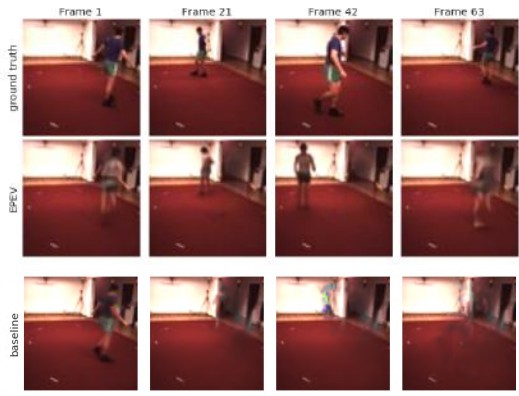

Figure 6: A visual comparison of the EPEV method and CDNA from Finn et al. (2016) as the baseline. This example is cherry picked to show results when there is significant movement in the ground truth. See appendix section G for non cherry picked results. The contrast of these images was increased to make the humans easier to see. In CDNA from Finn et al. (2016), the person disappeared part way through the prediction. The EPEV method, produced relatively sharp predictions up until frame 42, and a blurry human prediction at frame 63.

From visually inspecting the images we found that in images where there is not significant movement in the first 5 frames of the ground truth it is hard to tell the difference between our method and CDNA

since both methods predict an image similar to the early ground truth frames. However, when there is significant movement in the first 5 ground truth frames, the predictions from EPEV are sharper further into the future than CDNA. See the appendix section G for images where there is significant movement in the first 5 ground truth frames so the methods can be compared. We also collected results from the E2E method, but those blur out very quickly and are shown in appendix section G.

The CDNA method from Finn et al. (2016) produces blurry images since it is trained to minimize L2 loss directly (Finn et al., 2016). In the EPEV method, the predictor and VAN are trained separately. This prevents the VAN from learning to produce blurry images when the predictor is not confident. The predictions will be sharp as long as the predictor network predicts a valid encoding.

We also compare our method to Villegas et al. (2017). This method gives sharper results than ours. We think that this is because Villegas et al. (2017) uses a adversarial loss (Goodfellow et al., 2014) and since nothing besides the human is moving in this dataset, the pose works well as a high level structure.

### 4.3.1 PERSON DETECTOR EVALUATION

We propose to compare the methods quantitatively by considering whether the generated videos contain a recognizable person. To do this in an automated fashion, for each of the generated frames, we ran a MobileNet (Howard et al., 2017) object detection model pretrained on the MS-COCO (Lin et al., 2014) dataset. We recorded how confident the detector was that a person (one of the MS-COCO labels) is in the image. We call this the "person score" (its value ranges from 0 to 1, with a higher score corresponding to a higher confidence level). The results on each frame averaged over 1k runs are shown in Figure 7.

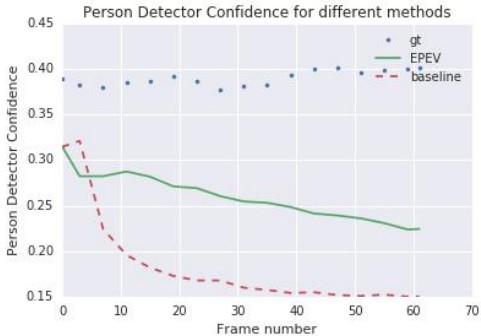

Figure 7: Confidence of the person detector that a person is in the image ("person score"). The baseline method is CDNA from Finn et al. (2016).

The person score on the ground truth frames is about 0.4. This is likely due to the mismatch between the training set images of the model (the MS-COCO dataset images are very different in terms of image statistics compared to the Humans 3.6M data). The person score is 0.26 on average for the images generated by the EPEV method, and 0.18 for CDNA from Finn et al. (2016). The person score degrades very rapidly in the first 8 frames of CDNA, but degrades more slowly in the EPEV method. The person score of the EPEV method on frame 63 is about the same as on frame 8 of CDNA. This confirms our visual analysis that the EPEV method produces clearer predictions further into the future. The EPEV method was only trained to predict 32 frames into the future but there is no significant drop in the person score at frame 32, showing that the EPEV method generalizes well to predicting longer sequences.

### 4.3.2 HUMAN EVALUATION

We also used a service similar to Mechanical Turk to collect comparisons of 1,000 generated videos from our EPEV method and the CDNA baseline. The task showed videos generated by the two methods side by side and asked raters to confirm whether one of the videos is more realistic. The workers rated the EPEV method as more realistic 53.6% of the time, the CDNA method as more realistic 11.1% of the time and the videos as being about the same 35.3% of the time. The high

number of "same" responses could be because of it being difficult to tell the difference between the methods when there is little movement.

## 5 CONCLUSION AND FUTURE WORK

On datasets where the pose does not capture all of the information needed to predict future frames, letting the network define its own high level structure in addition to the pose is an improvement upon a Villegas et al. (2017). The EPEV method generates sharper images than Finn et al. (2016) on non deterministic datasets, and can generate further into the future on a toy dataset that we introduced. We posit an adversarial loss between the predictor and encoder would likely help with potentially uncertain scenarios and would fix the problem of the EPEV method sometimes generating blurry images,

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

# Appendices

## A    IMPLEMENTATION DETAILS

The convolutional neural net encoder is a VGG-16 network (Simonyan & Zisserman, 2014). In the EPEV method, the encoder needs to be pre-trained on Imagenet (Deng et al., 2009). If that is not done, the VAN will ignore the output from the encoder. All other configurations are run without pretraining.

The predictor network is a 20 layer LSTM, while the analogy network uses the deep analogy transformation described in Reed et al. (2015). The VAN outputs a mask which controls whether to use its own output, or the pixels of the first frame. This allows the network to focus on learning the changing parts of the image instead of the background.

The end effector orientation is converted to quaternions (Bousmalis et al., 2016) in order to calculate the loss if the pose is used. Layer normalization (Lei Ba et al., 2016) is used between every other layer in the VAN and predictor networks. We used dropout (Srivastava et al., 2014) on the encoder and VAN to prevent overfitting. The network overfits less in the INDIVIDUAL method, so we used less dropout.

## B    HUMAN EVALUATION DETAILS

Each video in the comparison is generated from the same starting sequence. The side that the EPEV method and the CDNA method are displayed on is changed randomly.

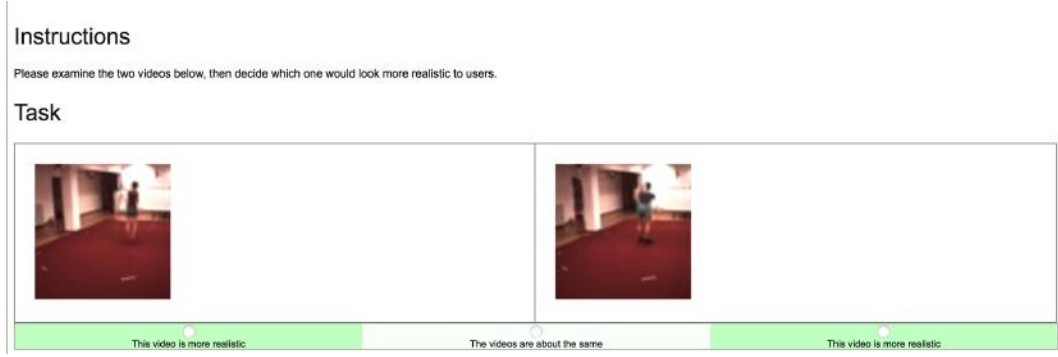

Figure 8: The screen shown to workers in the human evaluation

## C  EPEV METHOD WITH ENCODER OUTPUT FED TO VAN

To see what the encoder has learned in the EPEV method, we can obtain results from the visual analogy network given the input from the encoder. The encoder is given the ground truth image. The results are shown in figure 9. The results show that the encoder encodes where the person is

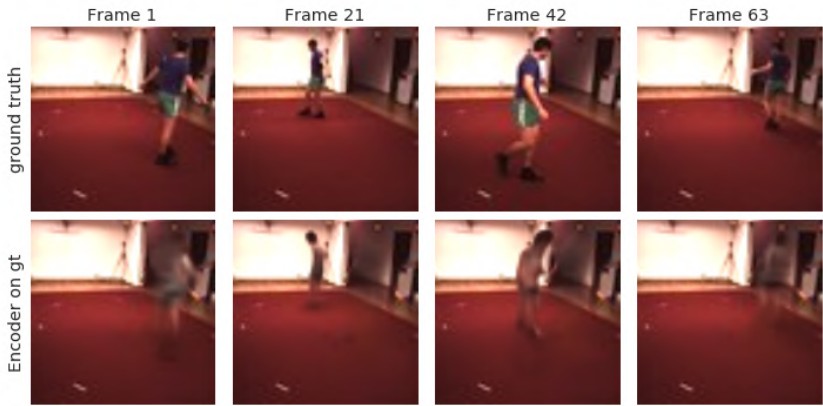

Figure 9: Results from the EPEV approach when the VAN is given the output of the encoder on the ground truth frame.

in the image, as well as the orientation of the arms, legs and head to some extent. The results are not as good as one would expect from an autoencoder, since the encoder has the constraint that the encoding also has to be easy to predict.

## D  TRAINING DETAILS

We trained all of the methods including Finn et al. (2016) for 3 million steps using async SGD, across 32 worker machines. We used a minibatch size of 8 sequences in each step. The minibatch size could be so small because there were multiple frames per sequence. In methods with multiple optimizers, a step is defined as running each optimizer once. The hyperparameters are optimized separately for both datasets on a validation set. We used the best learning rates for each method.

# E    MORE RESULTS ON SHAPES DATAEST.

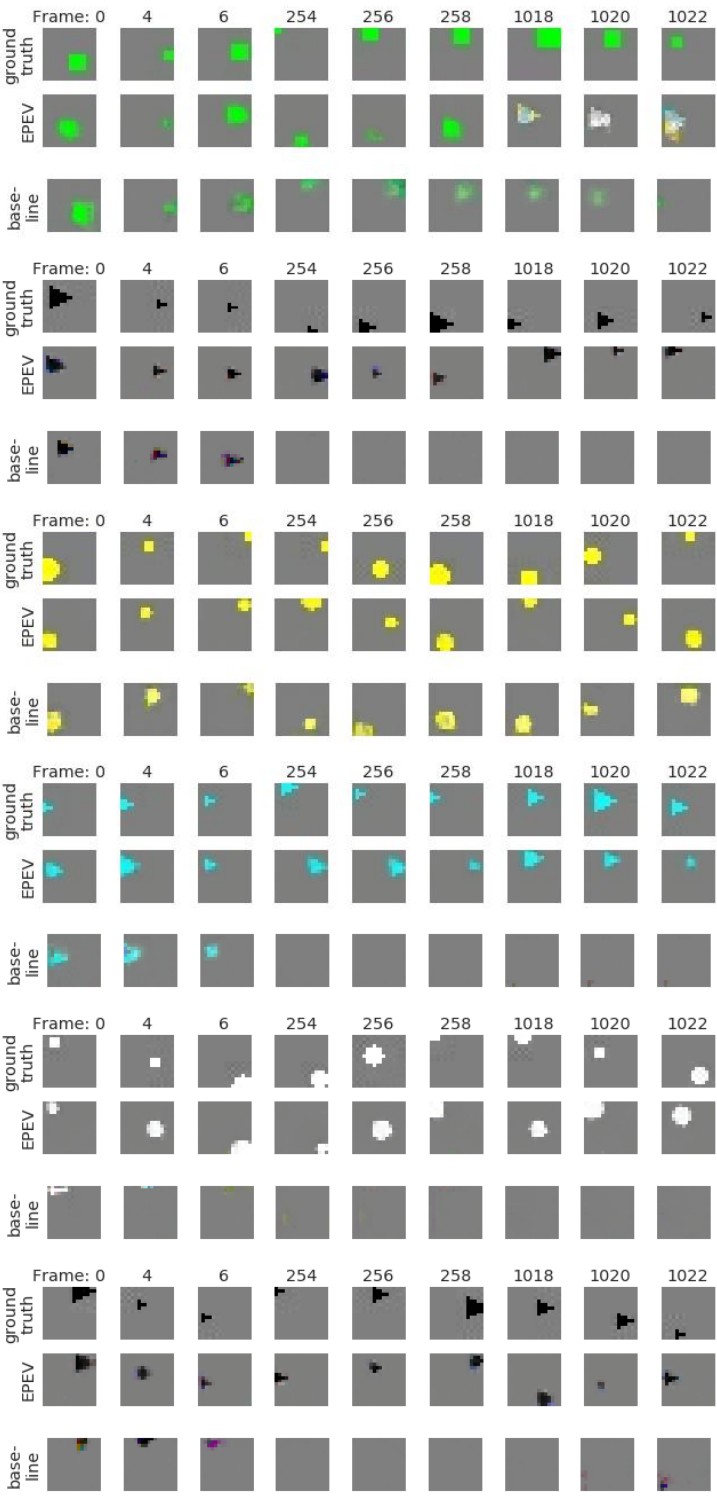

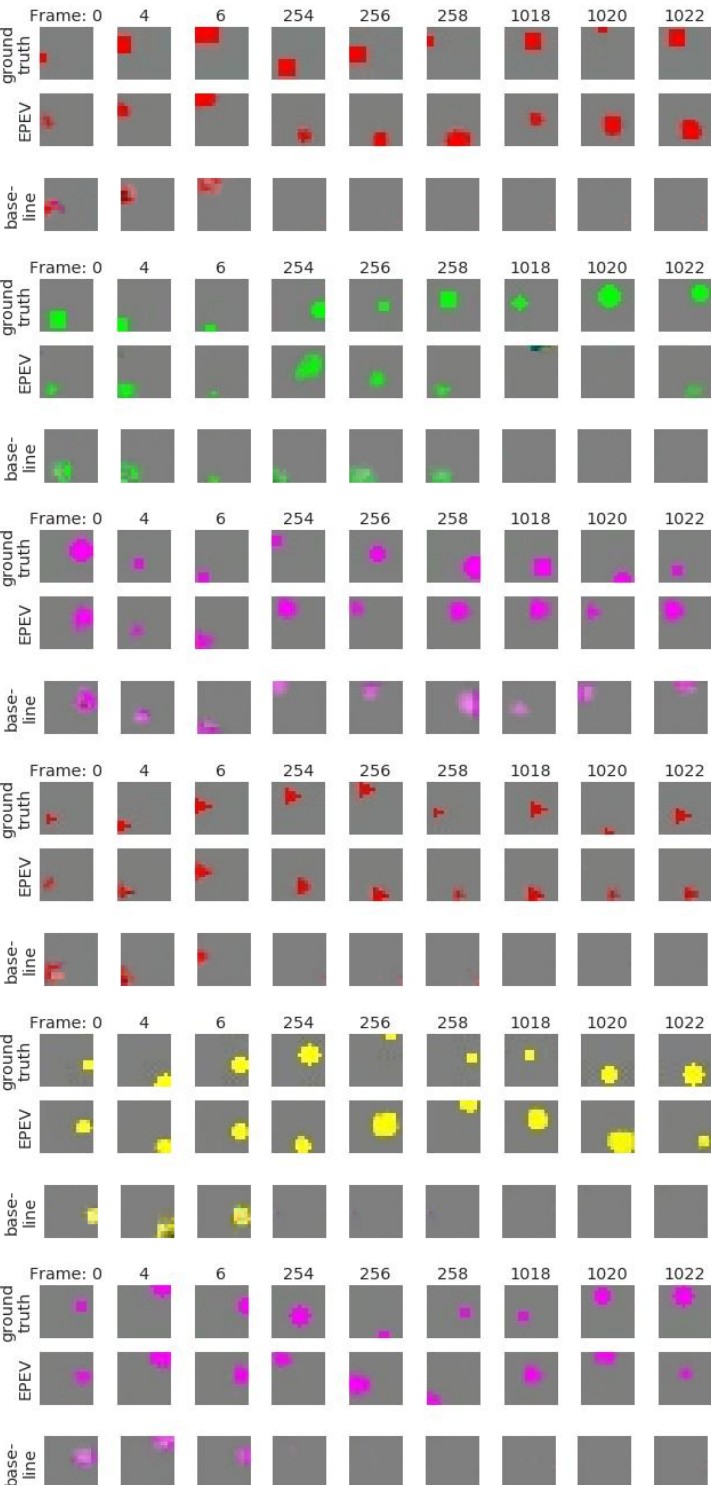

# F   MORE RESULTS ON ROBOT PUSH DATASET.

## F.1   FRAMES WHERE EPEV PREDICTS OBJECTS BETTER THAN INDIVIDUAL.

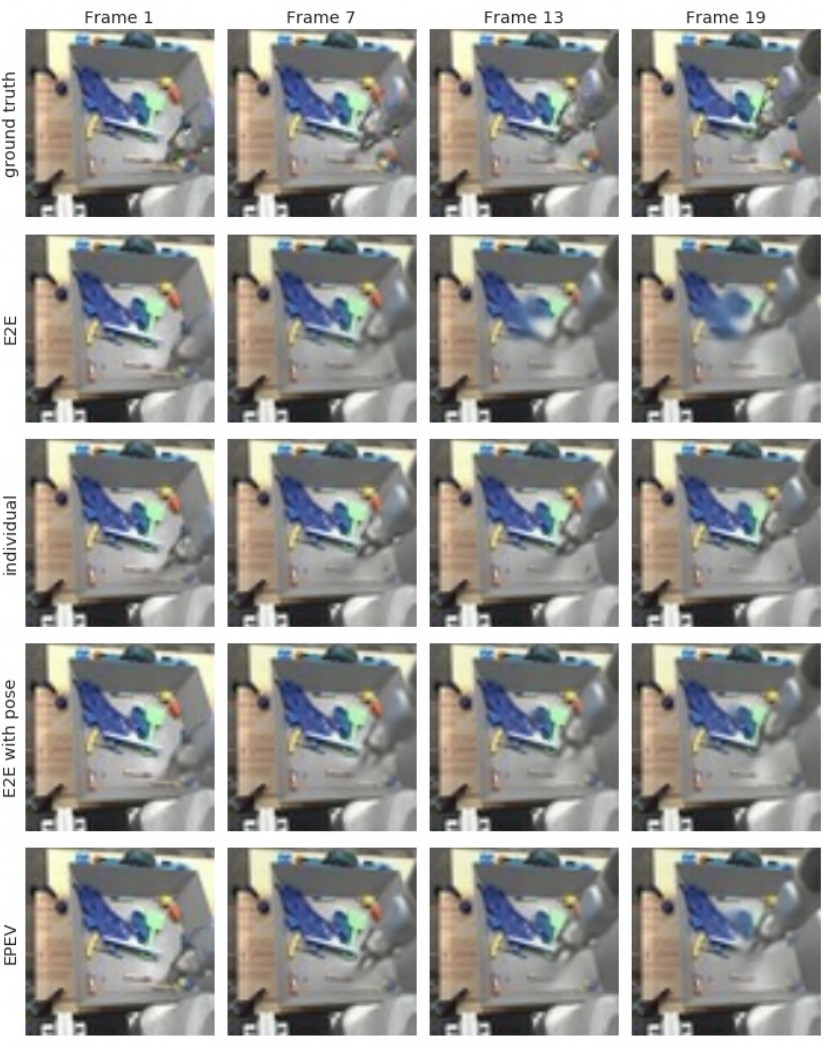

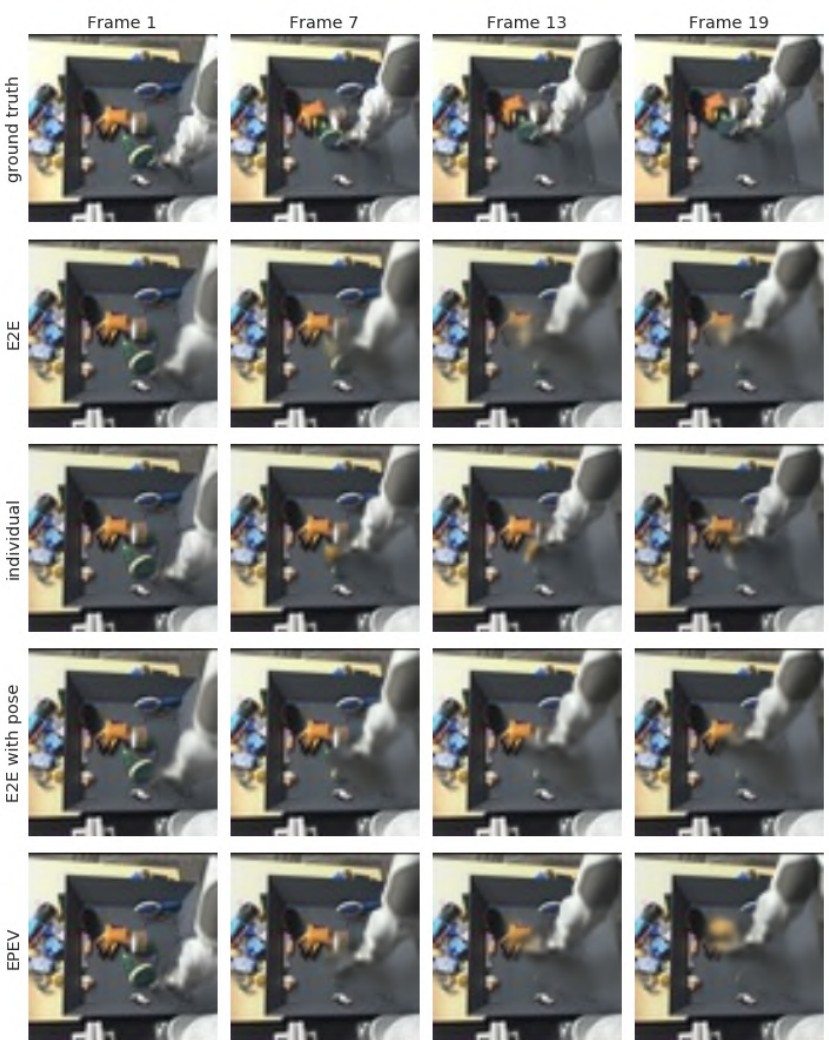

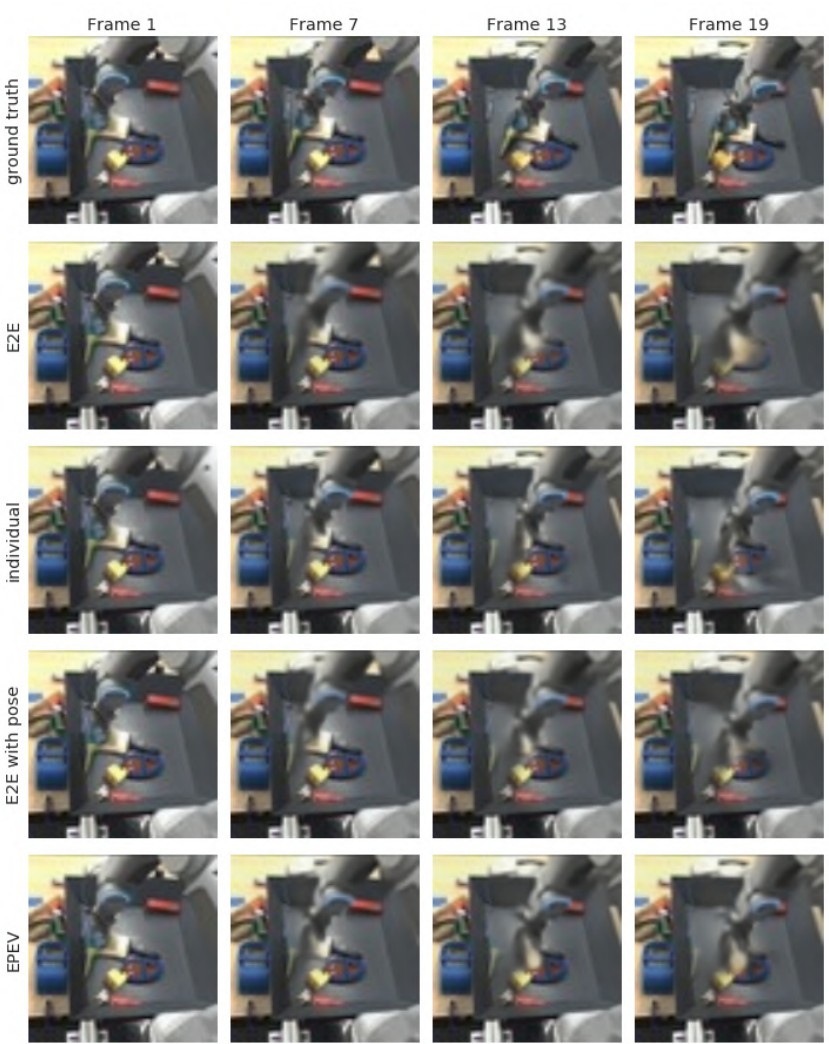

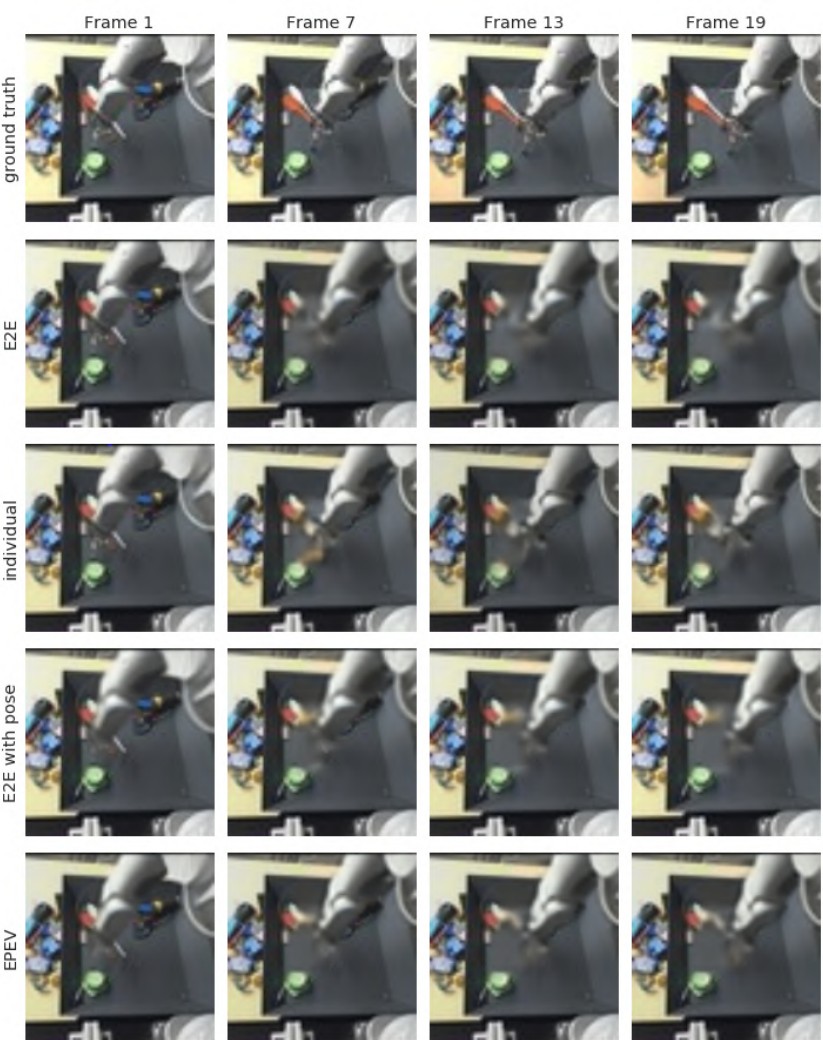

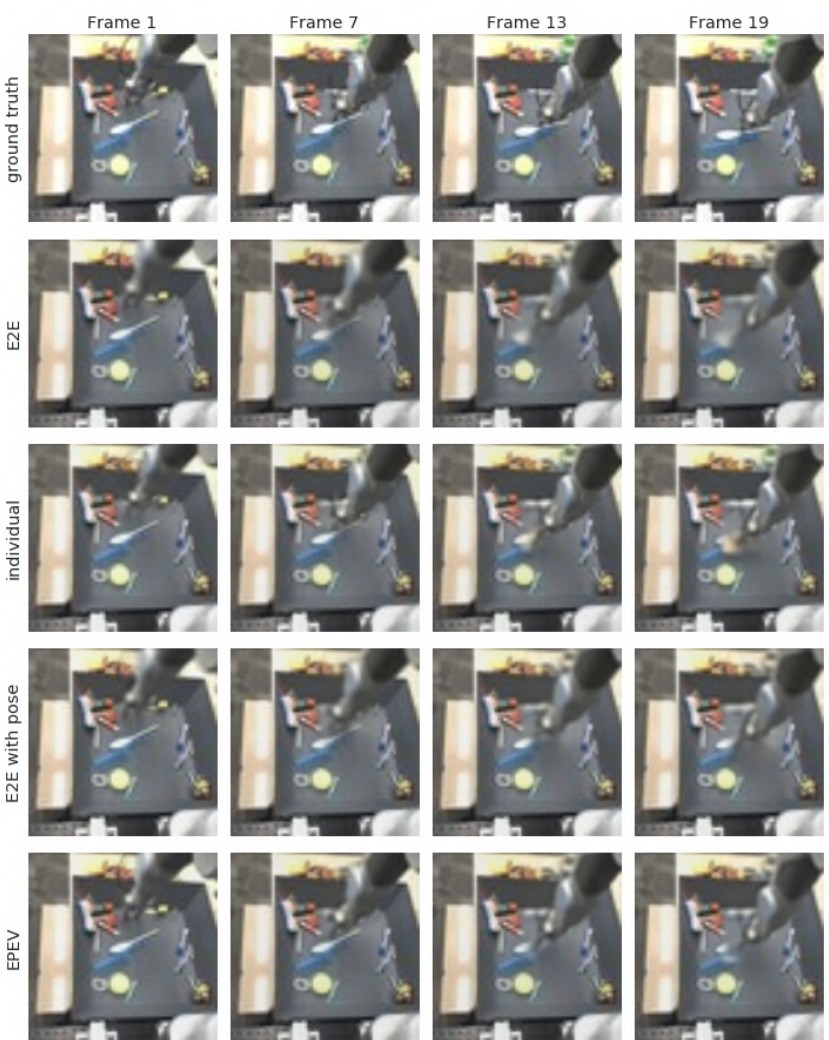

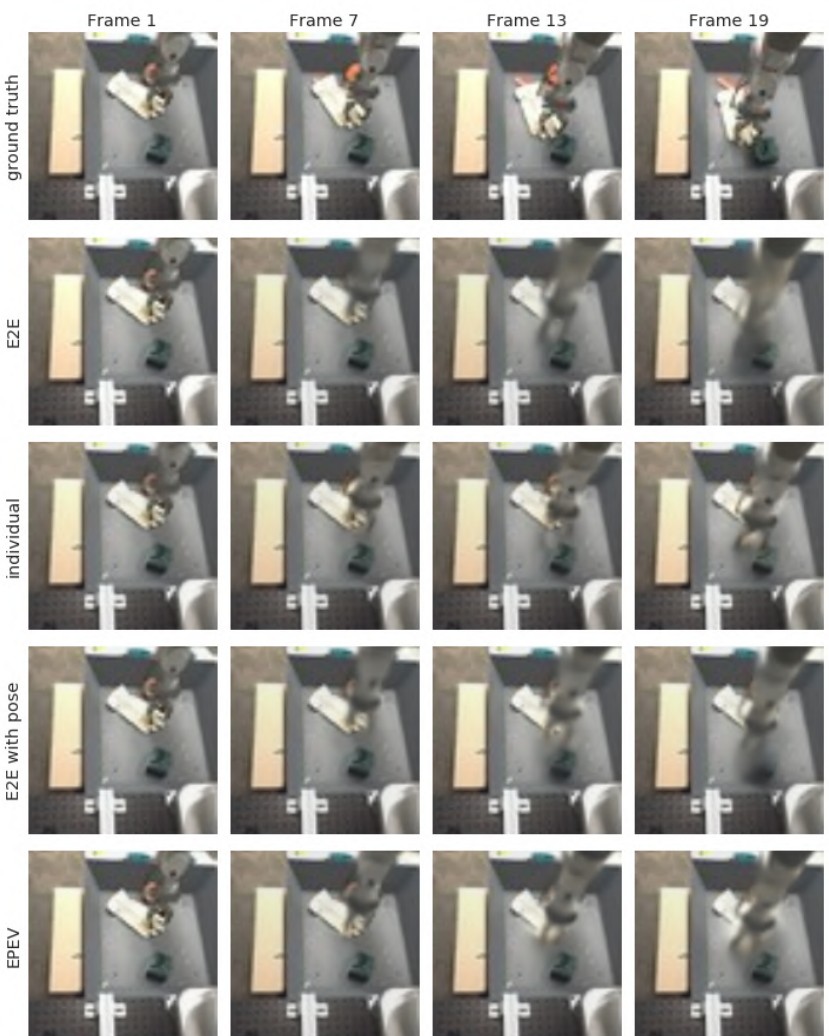

## F.2 FRAMES WHERE INDIVIDUAL PREDICTS OBJECTS BETTER THAN EPEV.

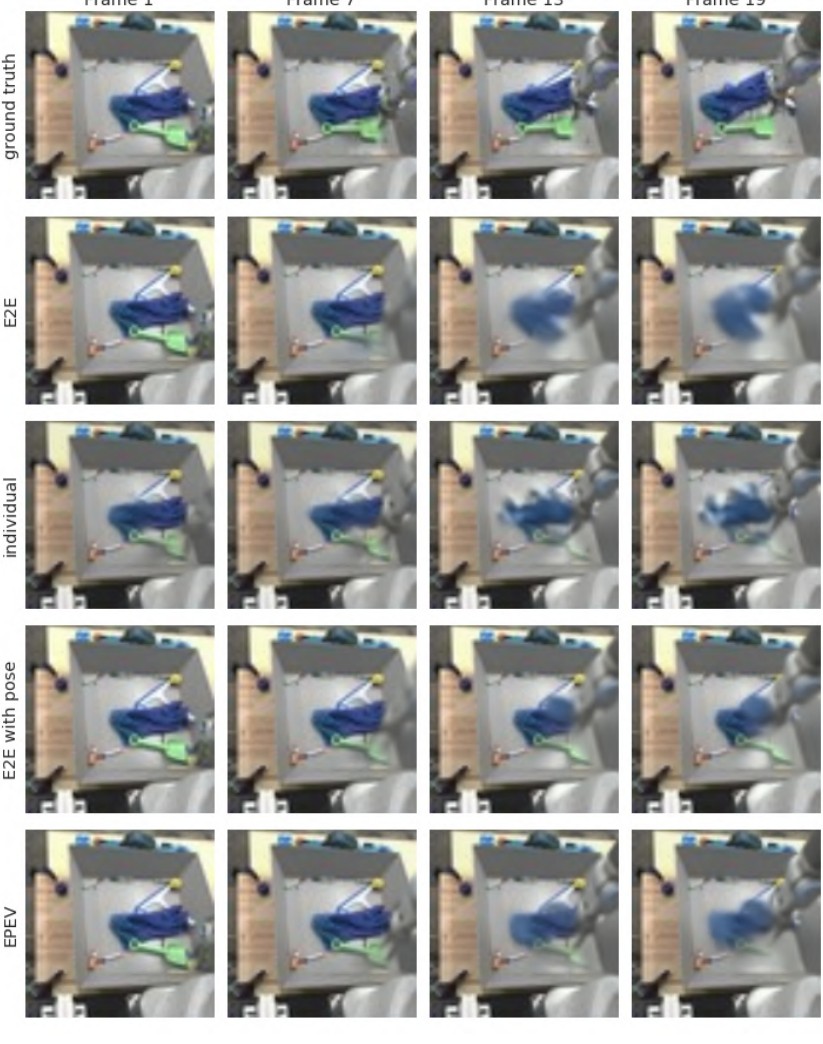

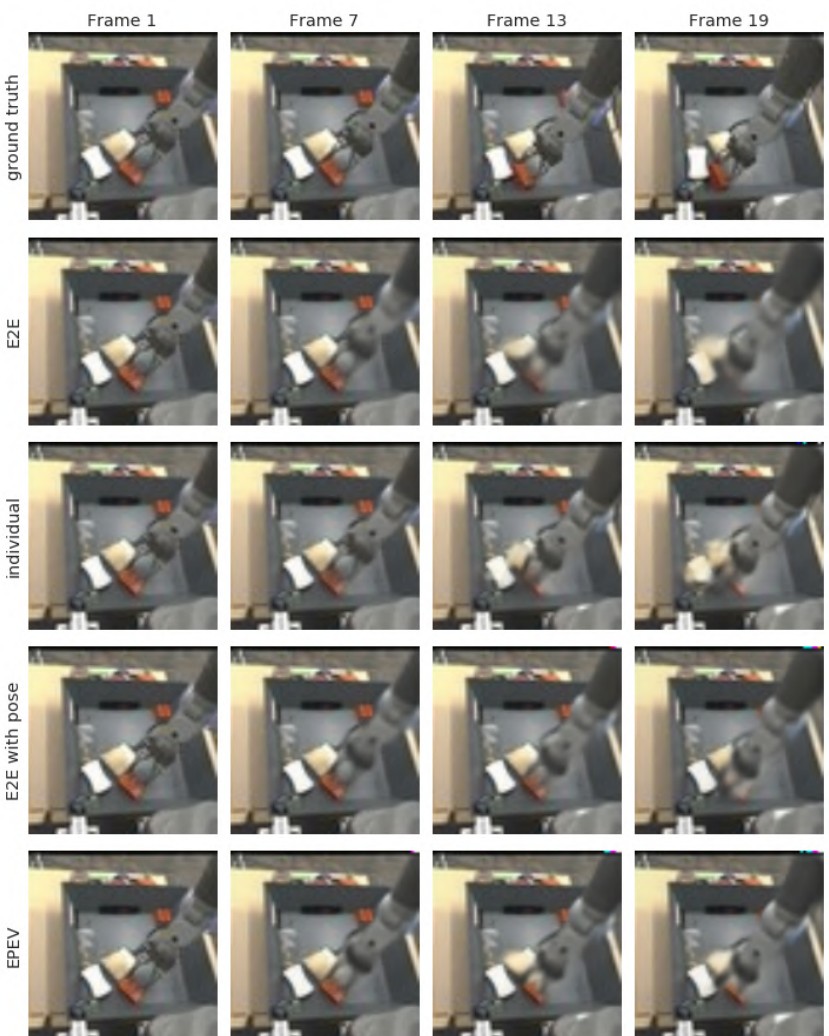

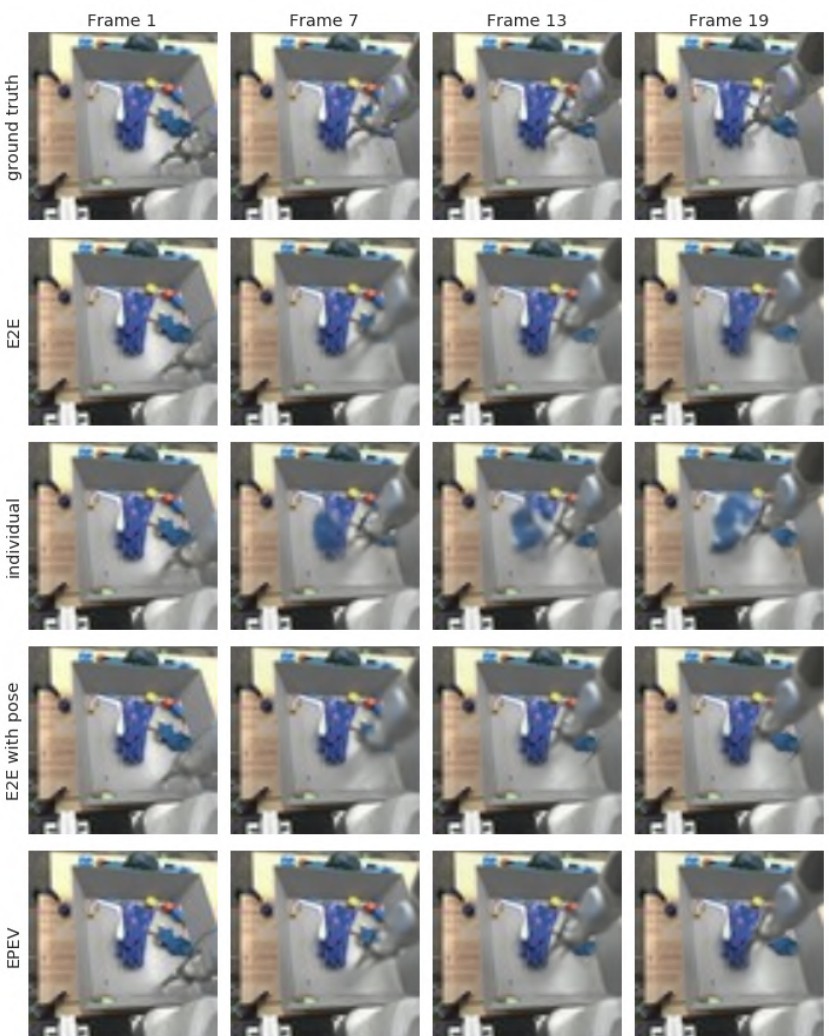

### F.3 FRAMES WHERE EPEV AND INDIVIDUAL PREDICT OBJECTS WITH THE SAME QUALITY.

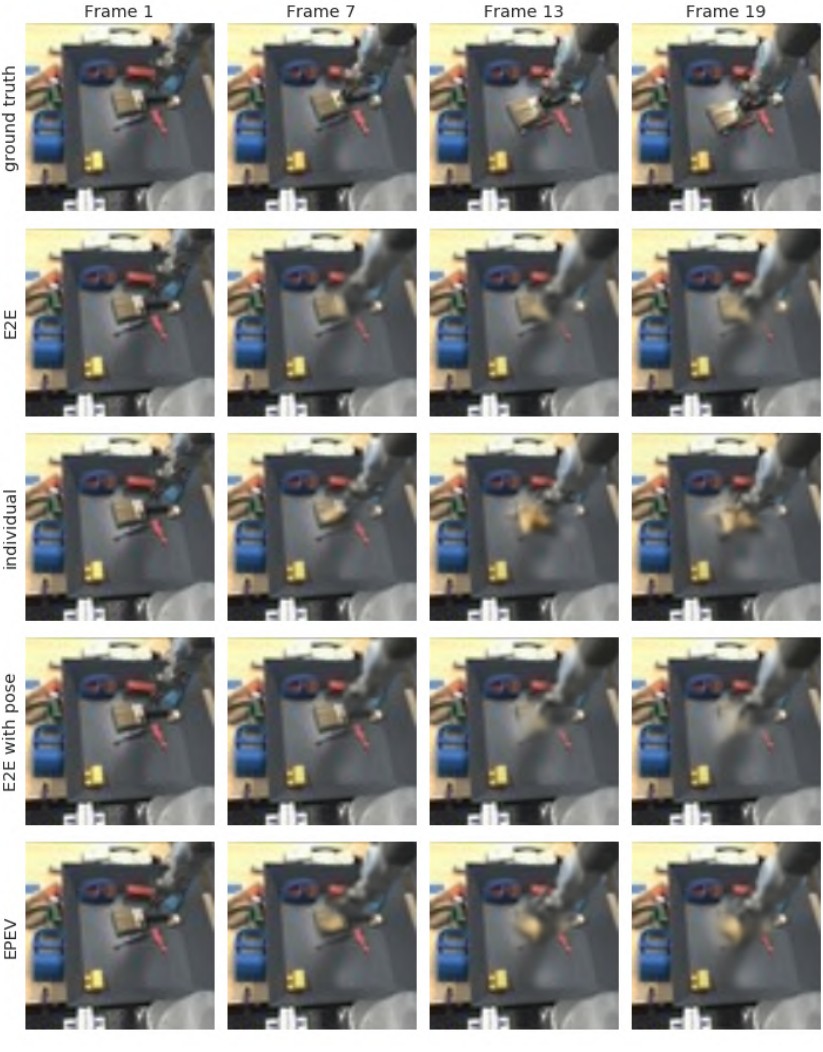

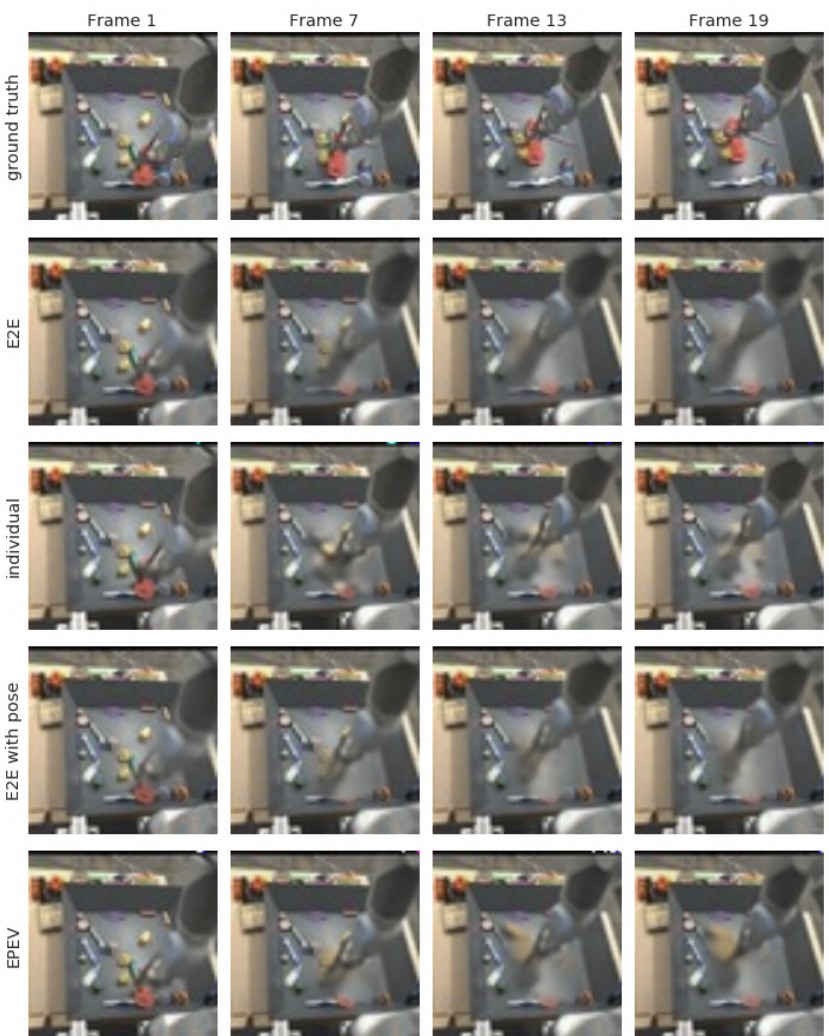

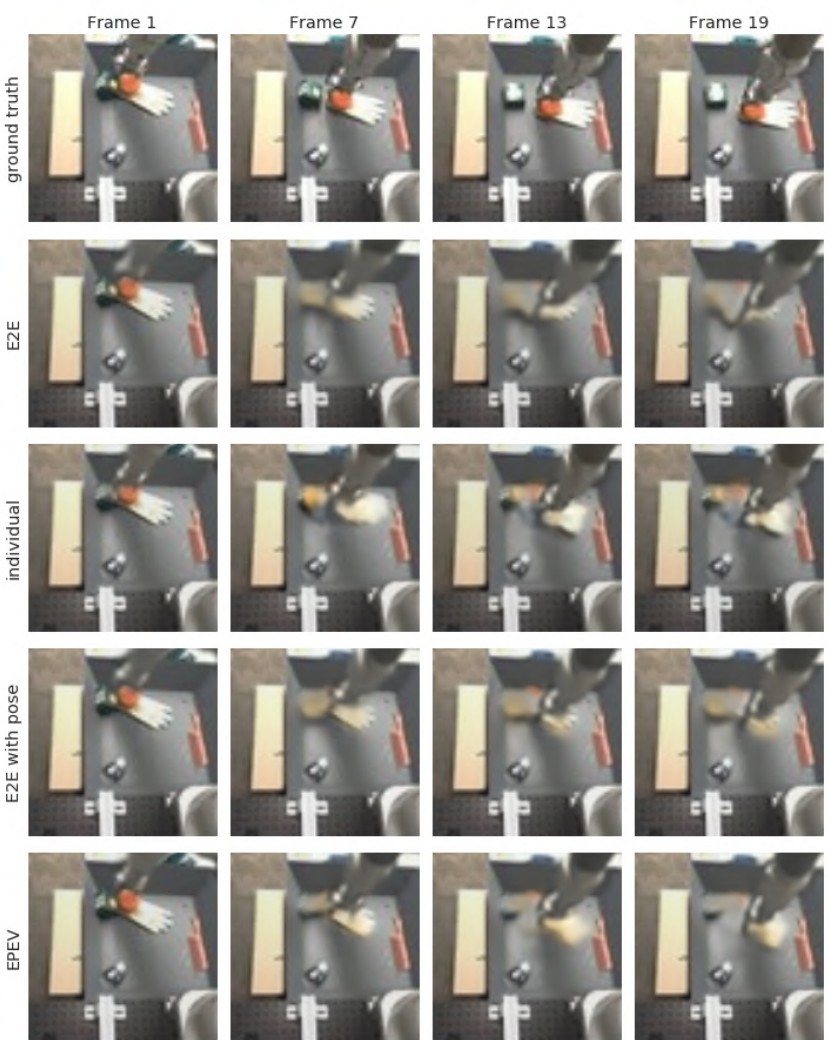

# G   MORE RESULTS ON HUMANS DATAEST.

## G.1   SIGNIFICANT MOVEMENT IN THE FIRST 5 GROUND TRUTH FRAMES.

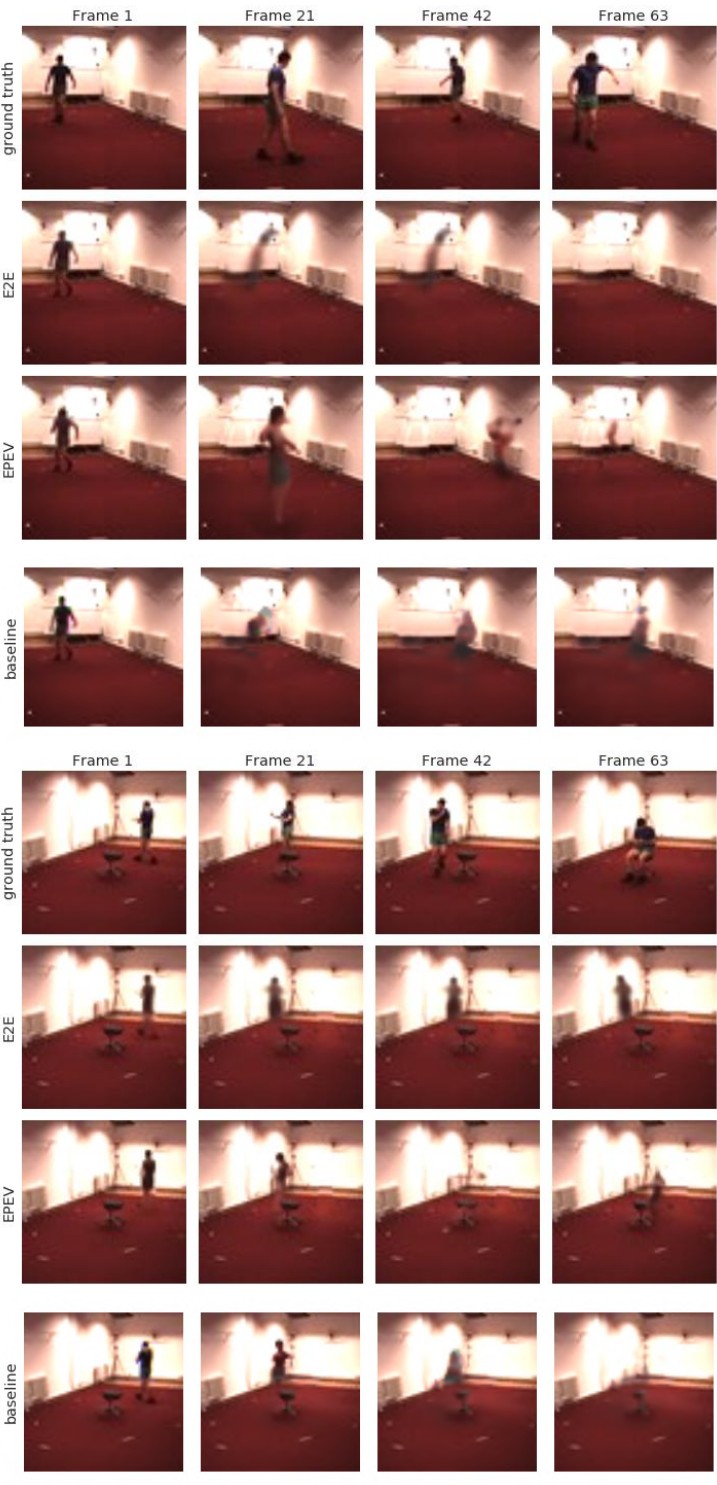

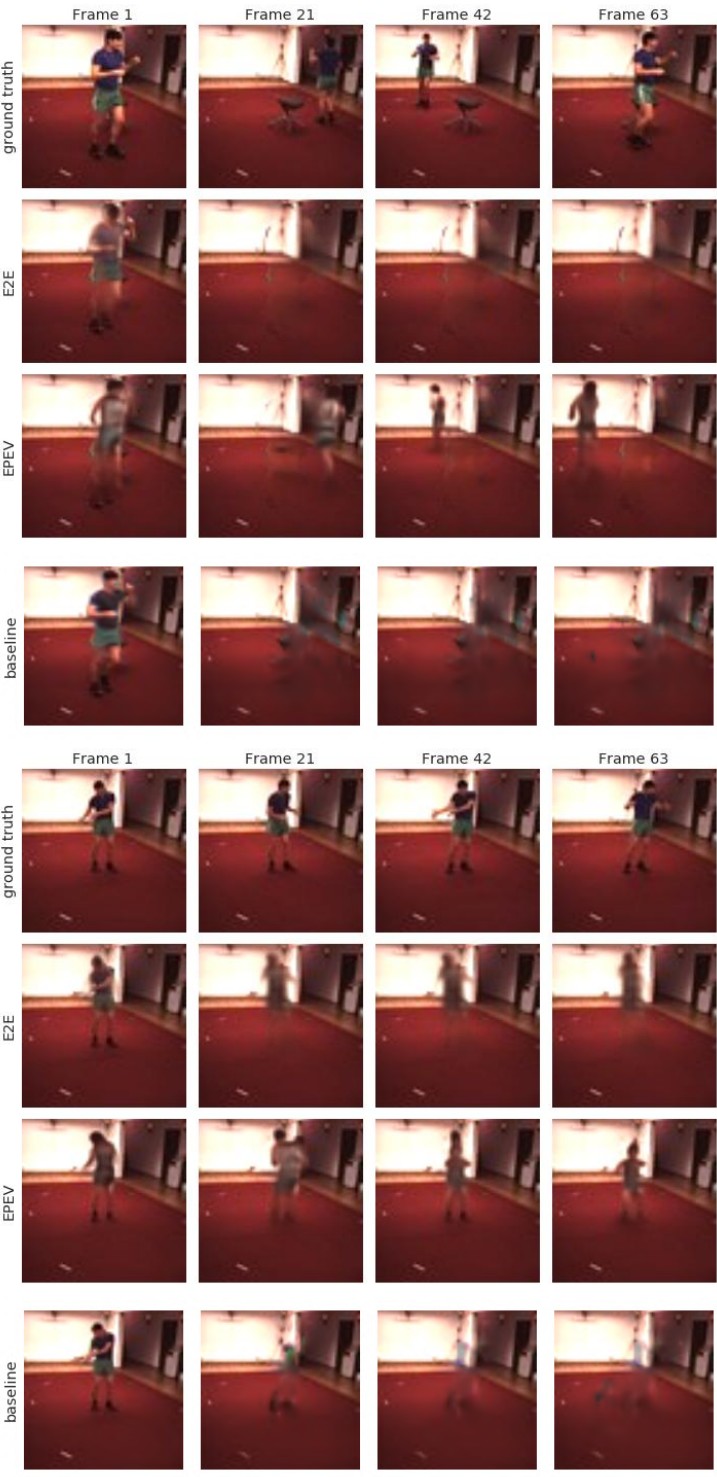

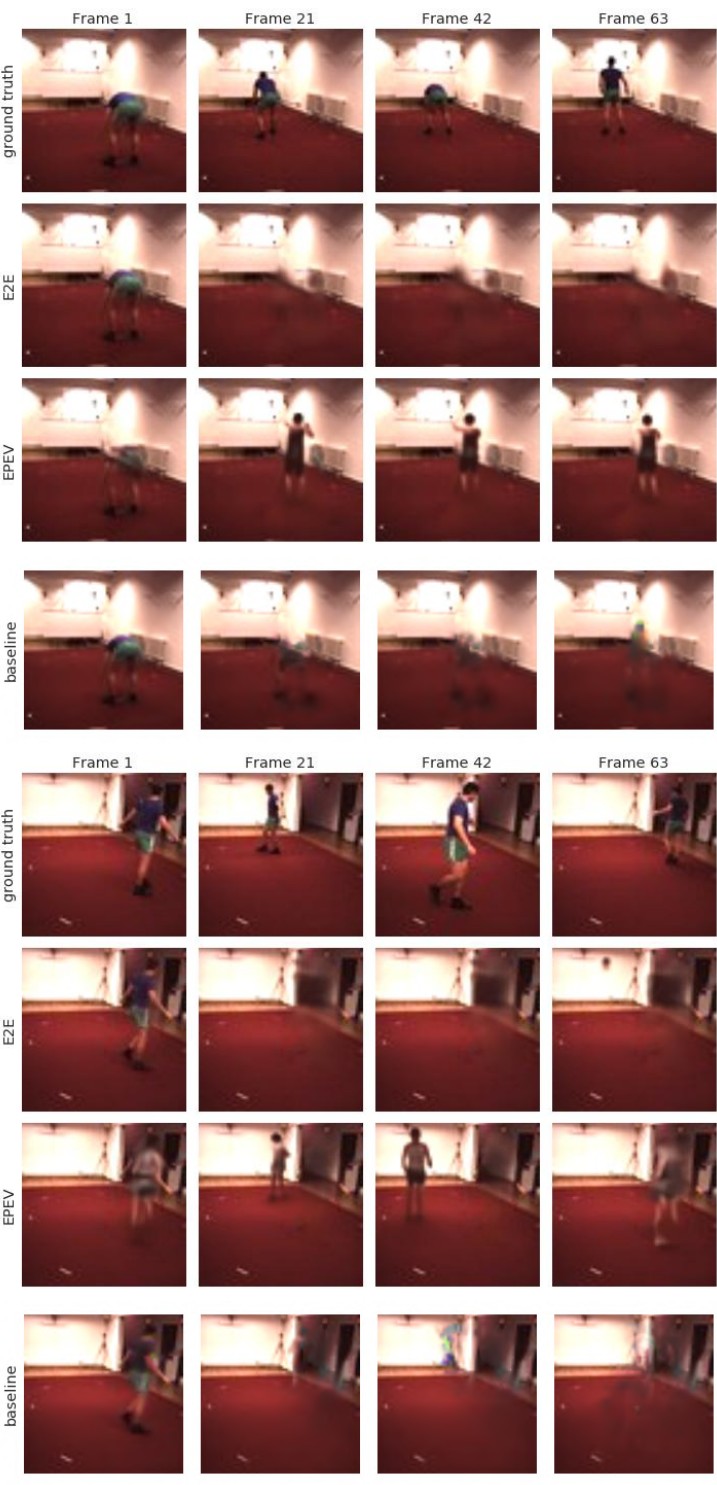

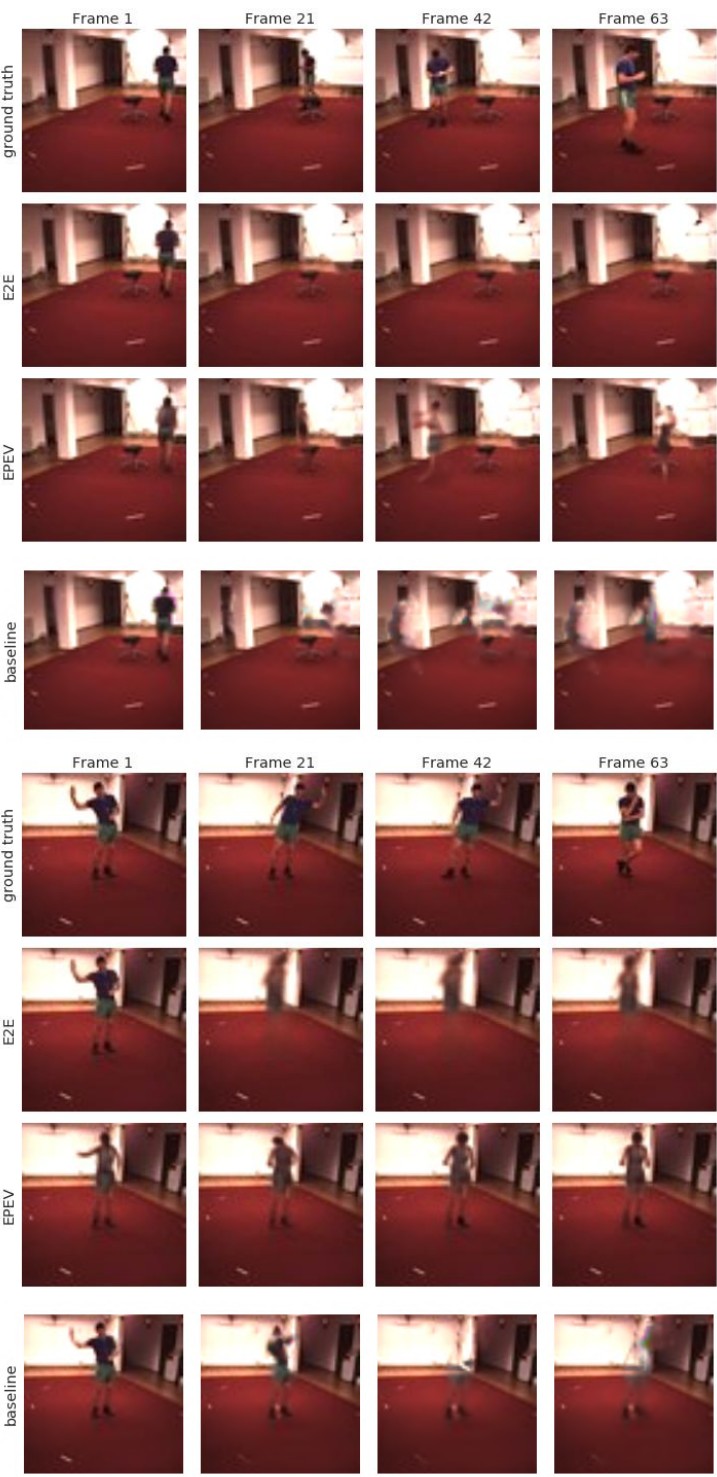

## G.2  NO SIGNIFICANT MOVEMENT IN THE FIRST 5 GROUND TRUTH FRAMES.

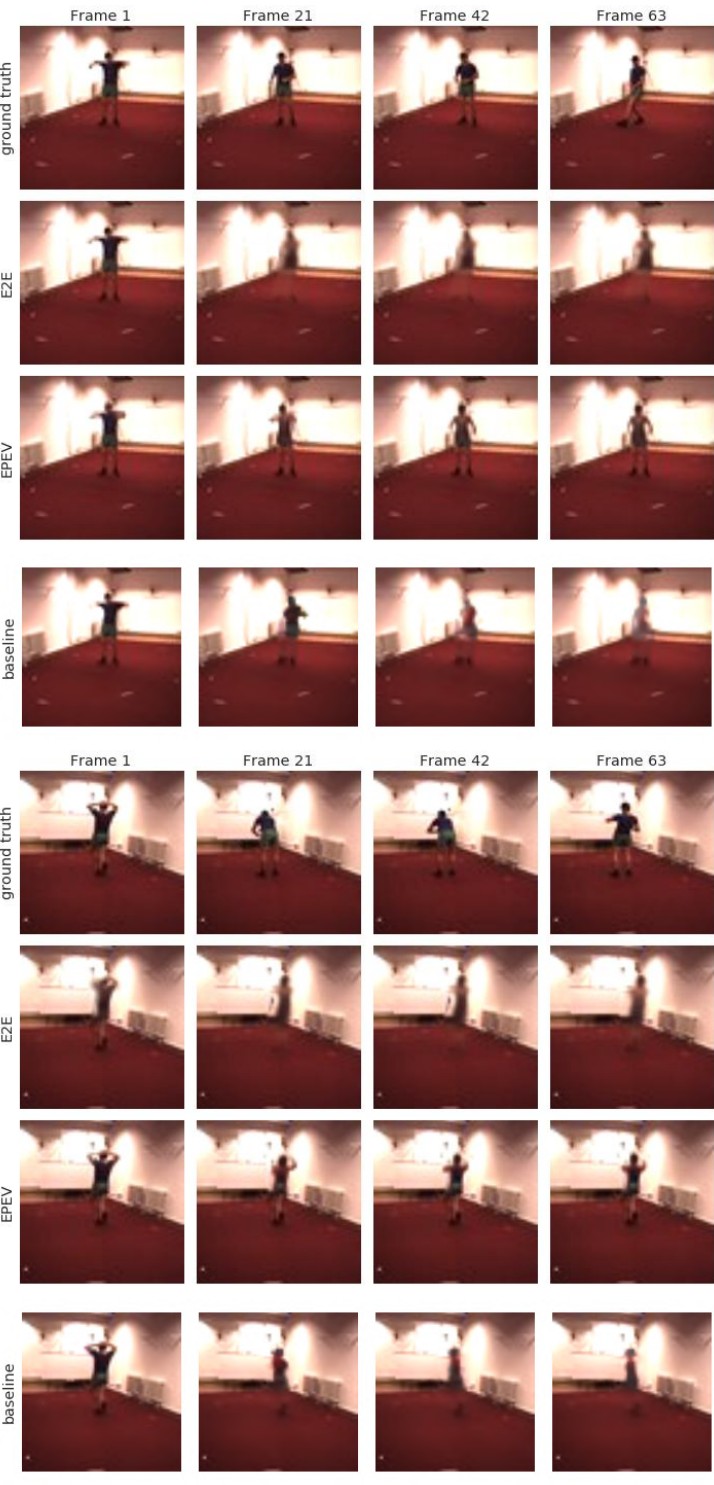

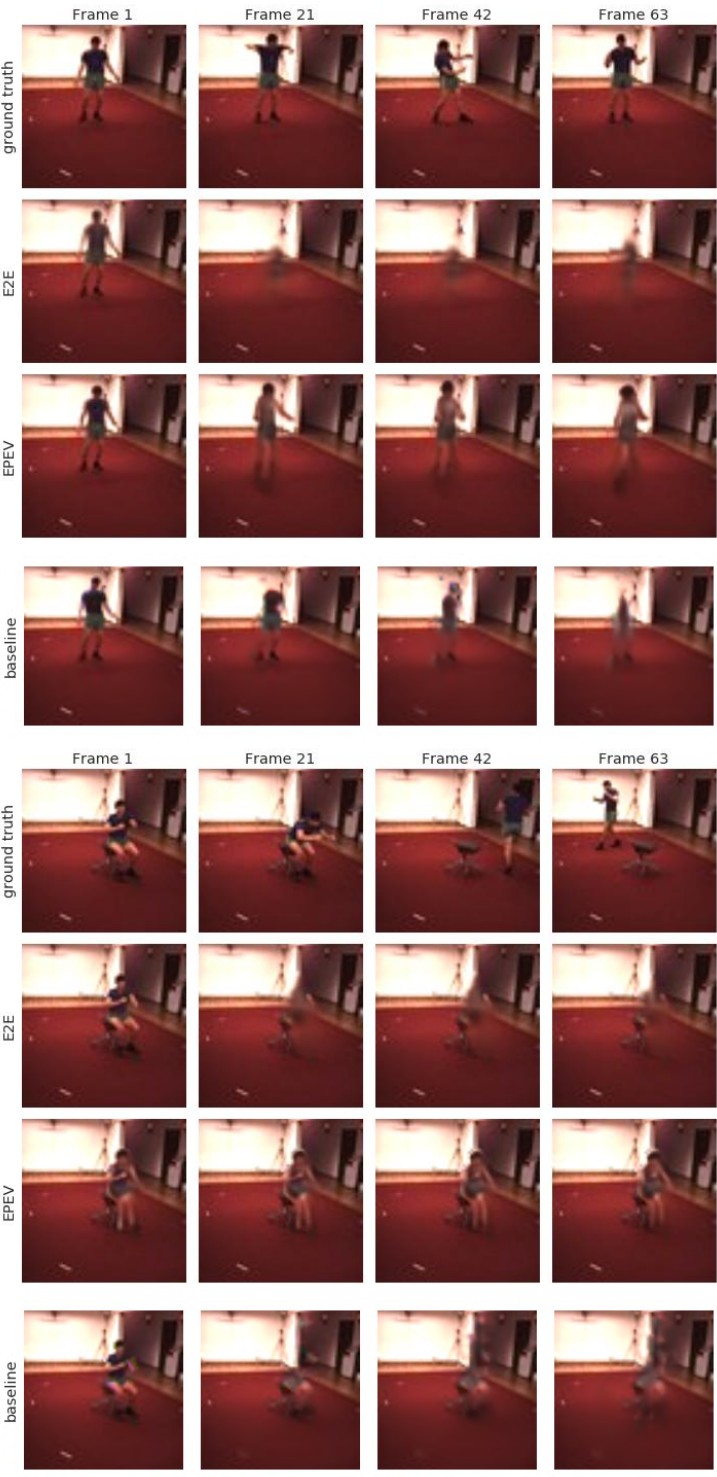

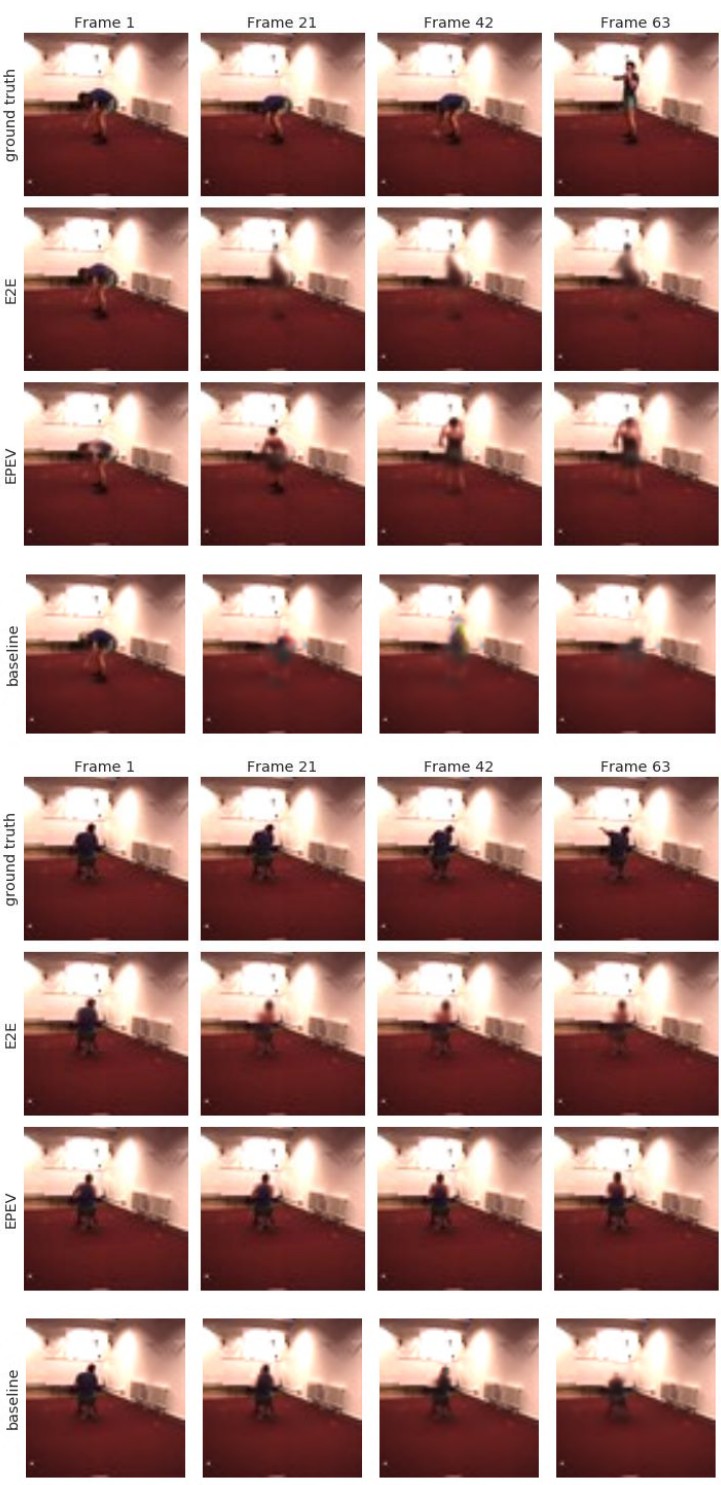

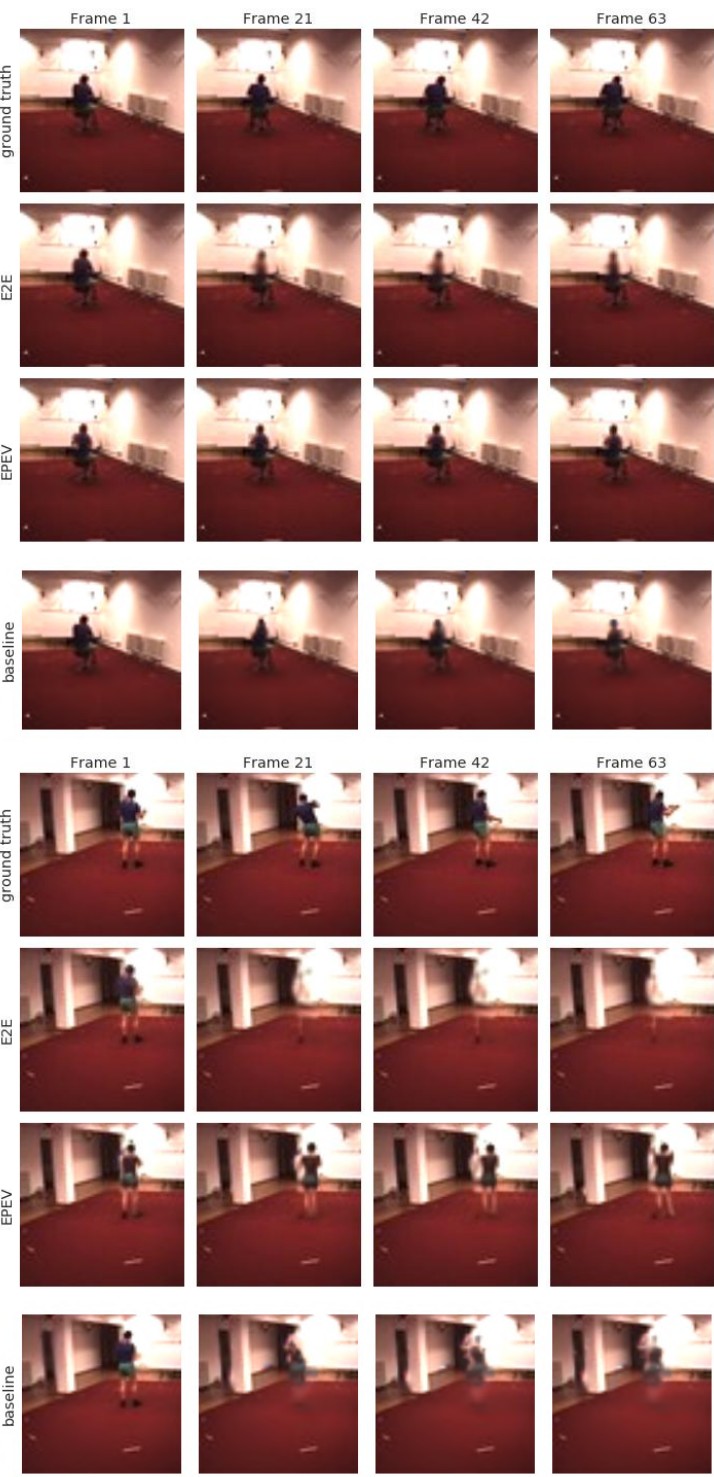

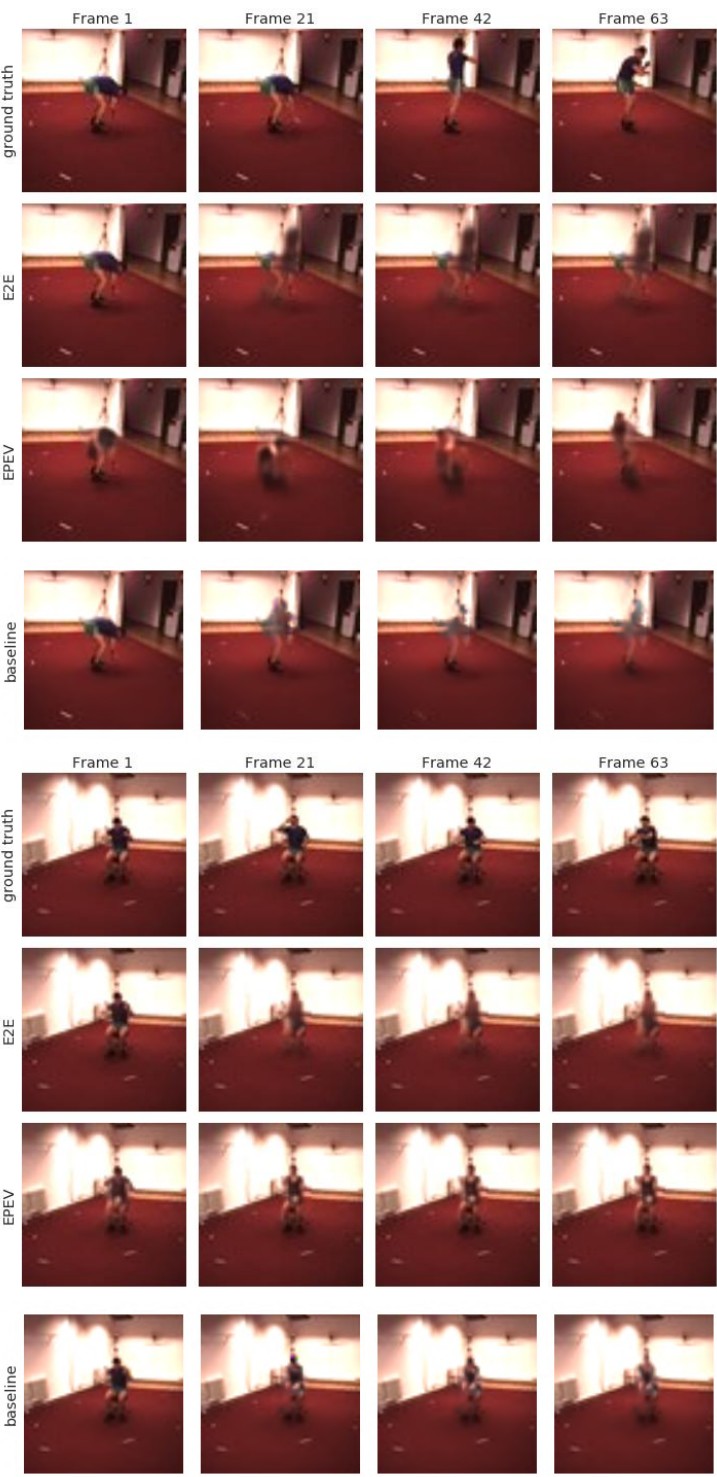

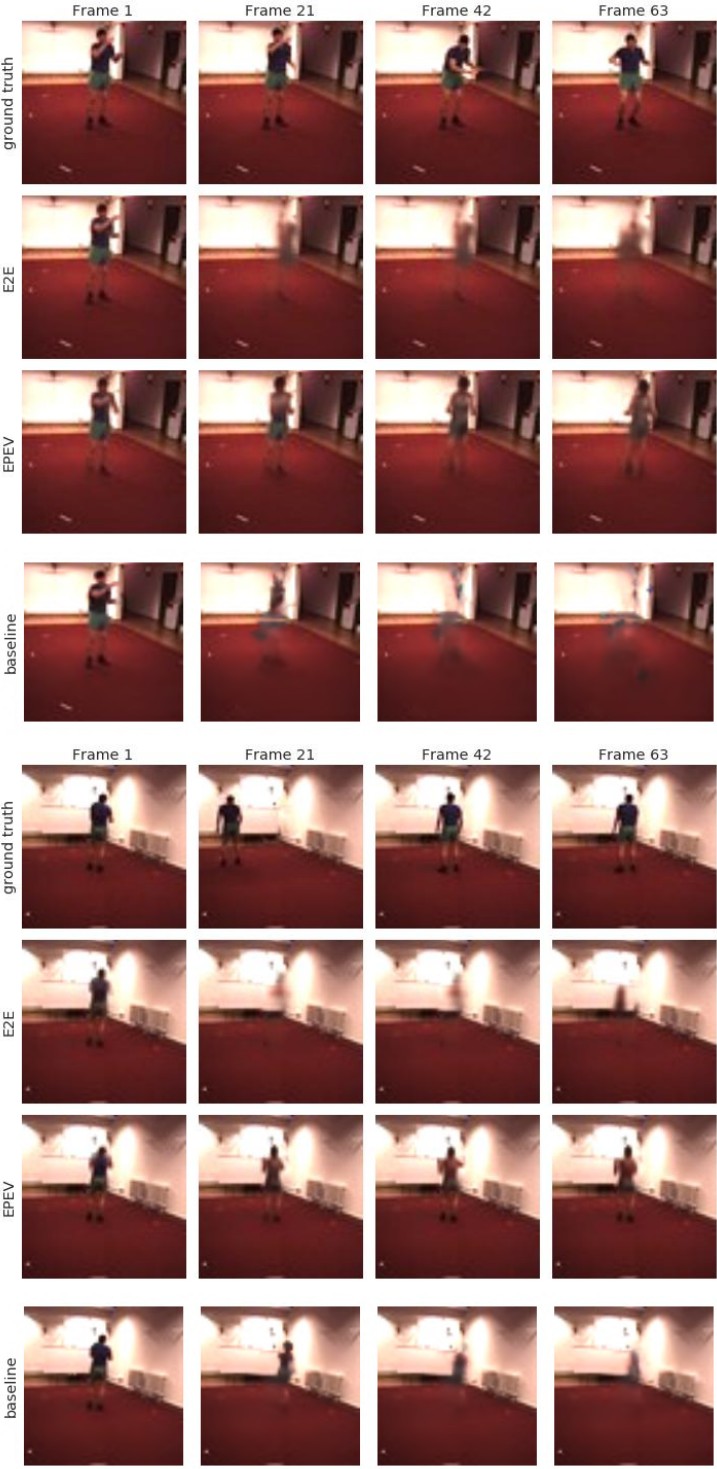

