# OpenReview forum: "Unsupervised Hierarchical Video Prediction"
_ICLR.cc/2018/Conference — Reject_

### Official Review · AnonReviewer1 · 2017-11-27
**Interesting paper but I find the claims are not backed up by the experimental evidence**

**Rating:** 4
**Confidence:** 4

**Review:**

The paper treats the interesting problem of long term video prediction in complex video streams. I think the approach of adding more structure to their representation before making longer term prediction is also a reasonable one. Their approach combines an RNN that predicts an encoding of scene and then generating an image prediction using a VAN (Reed et al.). They show some results on the Human3.6M and the Robot Push dataset.

I find the submission lacking clarity in many places. The main lack of clarity source I think is about what the contribution is. There are sparse mentions in the introduction but I think it would be much more forceful and clear if they would present VAN or Villegas et al method separately and then put the pieces together for their method in a separate section. This would allow the author to clearly delineate their contribution and maybe why those choices were made. Also the use of hierarchical is non-standard and leads to confusion I recommend maybe "semantical" or better "latent structured" instead. Smaller ambiguities in wording are also in the paper : e.g. related work -> long term prediction "in this work" refers to the work mentioned but could as well be the work that they are presenting.

I find some of the claims not clearly backed by a thorough evaluation and analysis. Claiming to be able to produce encodings of scenes that work well at predicting many steps into the future is a very strong claim. I find the few images provided very little evidence for that fact. I think a toy example where this is clearly the case because we know exactly the factors of variations and they are inferred by the algorithm automatically or some better ones are discovered by the algorithm, that would make it a very strong submission. Reed et al. have a few examples that could be adapted to this setting and the resulting representation, analyzed appropriately, would shed some light into whether this is the right approach for long term video prediction and what are the nobs that should be tweaked in this system.

In the current format, I think that the authors are on a good path and I hope my suggestions will help them improve their submission, but as it stands I recommend rejection from this conference.

---

### Official Review · AnonReviewer3 · 2017-11-28
**interesting architecture alternatives, but lack of strong empirical conclusions regarding the lack of supervising the high level structure**

**Rating:** 4
**Confidence:** 4

**Review:**

The paper presents a method for hierarchical future frame prediction in monocular videos. It builds upon the recent method of Villegas et al. 2017, which generates future RGB frames in two stages: in the first stage, it predicts a human body pose sequence, then it conditions on the pose sequence to predict RGB content, using an image analogy network. This current paper, does not constrain the first stage (high level) prediction to be human poses, but instead it can be any high level representation. Thus, the method does not require human annotations.

The method has the following two sub-networks:
1) An image encoder, that given an RGB image, predicts a deep feature encoding.
2) An LSTM predictor, that conditioned on the last observed frame's encoding,  predicts future high level structure p_t. Once enough frames are generated though, it conditions on its own predictions.
3) A visual analogy network (VAN), that given predicted high level structure p_t, it predicts the pixel image I_t, by applying the transformation from the first to tth frame, as computed by the vector subtraction of the corresponding high level encodings (2nd equation of the paper). VAN is trained to preserve parallelogram relationships in the joint RGB image and high level structure embedding.

The authors experiment with  many different neural network connectivities, e.g., not constraining the predicted high level structure to match the encoder's outputs, constraining the predicted high level structure to match the encoder's output (EPEV), and training together the VAN  and predictor so that VAN can tolerate mistakes of the predictor. Results are shown in H3.6m and the pushobject datasets, and are compared against the method of Villegas et all (INDIVIDUAL). The conclusion seems to be that not constraining the predicted high level structure to match the encoder’s output, but biasing the encoder’s output in the observed frames to represent ground-truth pose information, gives the best results.

Pros
1) Interesting alternative training schemes are tested

Cons:
1)Numerous English mistakes, e.g.,  ''an intelligent agents", ''we explore ways generate" etc.

2) Equations are not numbered (and thus is hard to refer to them.) E.g., i do not understand the first equation, shouldn’t it be that e_{t-1} is always fixed and equal to the encoding of the last observed (not predicted) frame? Then the subscript cannot be t-1.

3) In H3.6M, the results are only qualitative. The conclusions from the paper are uncertain, partly  due to the difficulty of evaluating the video prediction results.


Given the difficulty of assessing the experimental results quantitatively (one possibility to do so is asking a set of people of which one they think is the most plausible video completion), and given the limited novelty of the paper, though interesting alternative architectures are tried out, it may not be suitable to be part of  ICLR proceedings as a conference paper.

---

### Official Review · AnonReviewer2 · 2017-12-04
**insufficient novelty and significance**

**Rating:** 4
**Confidence:** 4

**Review:**

The paper presents a method for predicting future video frames. The method is based on Villegas et al. (2017), with the main difference being that no ground truth pose is needed to train the network.

The novelty of the method is limited. It seems that there is very little innovation in terms of network architecture compared to Villegas et al. The difference is mainly on how the network is trained. But it is straightforward to train the architecture of Villegas et al. without pose -- just use any standard choice of loss that compares the predicted frame versus the ground truth frame. I don't see what is non-trivial or difficult about not using pose ground truth in training.

Overall I think the contribution is not significant enough.

---

### Author Response · Authors · 2017-12-20
**Revised paper**

We would like to thank the reviewers for taking the time to read our submission and make numerous suggestions for improvements. We uploaded a revision that fixes some English mistakes, improves clarity, and provides new quantitative results on Humans 3.6M that highlight the benefits of the proposed method.

All three reviewers view the proposed method as insufficiently novel and/or significant. The main motivation of our work is to enable training of hierarchical video prediction models on data where pose groundtruth labels are impractical to collect or unavailable. As Reviewer2 points out, the simple way to do so would be to take the architecture of Villegas et al. and train it using a loss that compares the predicted frame versus the ground truth frame. This is essentially the E2E method described in our submission, and we include this model as one of the baselines in our work. In our experiments, we found that this E2E baseline is not sufficient, and we show that the EPEV method-- which is the main novel contribution of our work--produces much better results on the humans dataset than the E2E method, as shown in the appendix.

An important concern was that our submission did not have quantitative evidence to support our claims that our results on the Humans 3.6M dataset were better than the CDNA method from Finn et al. To address this, we ran the following evaluations:

1. We ran a pre trained person detector (ssd_mobilenet_v1_coco from the TensorFlow object detection model zoo) on the generated frames from the EPEV method and the CDNA method, on the Humans dataset. The person detector had a much higher average person-class confidence for EPEV frames, compared to CDNA. See section 4.3.1 of the most recent revision for details.

2. We also did a evaluation with a service similar to Mechanical Turk where workers rated the EPEV method as more realistic 53.6% of the time, the CDNA method as more realistic 11.1% of the time and the generated videos as being about the same 35.3% of the time. See section 4.3.2 of the most recent revision for details.

We believe this provides evidence that the proposed method is both qualitatively and quantitatively better compared to Finn et al. at predictions made on the Humans 3.6M dataset.

---

> ### Author Response · Authors · 2017-12-29
> **Revised paper Update**
>
> We also made the following revisions in a more recent update:
>
> 1. We seperated the description of the method from Villegas et al from the description of our method. We believe this addresses the comment from reviewer 1 about clarifying our key contribution. See section 2 and section 3.
>
> 2. We evaluated our method on a toy dataset and showed that it can make reasonable predictions 1,000 frames into the future about 97% of the time, where the CDNA baseline can only do this about 25% of the time. We believe this addresses the comment from reviewer 1 about needing evidence to support our claim that our method works well for long term prediction. See section 4.2.

---

### Decision · Program_Chairs · 2018-01-29
**ICLR 2018 Conference Acceptance Decision**

**Decision:**

Reject

**Comment:**

The paper presents a method for forward prediction in videos. The paper insufficiently motivates the proposed method and presents very limited empirical evaluations (no ablation studies, etc.) to backup its claims. This makes it difficult for the reader to put the work into  the context of the broader research around learning from unsupervised video data; leading reviewers to complete about perceived lack of novelty and clarity.